# Scale law of complex deformation transitions of nanotwins in stainless steel

A.Y. Chen [1], L.L. Zhu[2,3], L.G. Sun [4], J.B. Liu[3,5], H.T. Wang[2,3], X.Y. Wang[1], J.H. Yang[1] & J. Lu [4,6]

Understanding the deformation behavior of metallic materials containing nanotwins (NTs), which can enhance both strength and ductility, is useful for tailoring microstructures at the micro- and nano- scale to enhance mechanical properties. Here, we construct a clear deformation pattern of NTs in austenitic stainless steel by combining in situ tensile tests with a dislocation-based theoretical model and molecular dynamics simulations. Deformation NTs are observed in situ using a transmission electron microscope in different sample regions containing NTs with twin-lamella-spacing ($\lambda$) varying from a few nanometers to hundreds of nanometers. Two deformation transitions are found experimentally: from coactivated twinning/detwinning ($\lambda < 5\,\text{nm}$) to secondary twinning ($5\,\text{nm} < \lambda < 129\,\text{nm}$), and then to the dislocation glide ($\lambda > 129\,\text{nm}$). The simulation results are highly consistent with the observed strong $\lambda$-effect, and reveal the intrinsic transition mechanisms induced by partial dislocation slip.

[1] School of Materials Science and Engineering, University of Shanghai for Science and Technology, Shanghai 200093, China. [2] Department of Engineering Mechanics and Key Laboratory of Soft Machines and Smart Devices of Zhejiang Province, Zhejiang University, Hangzhou, Zhejiang 310027, China. [3] Center for X-Mechanics, Zhejiang University, Hangzhou, Zhejiang 310027, China. [4] Department of Mechanical Engineering, City University of Hong Kong, Hong Kong, China. [5] School of Materials Science and Engineering, Zhejiang University, Hangzhou, Zhejiang 310027, China. [6] Hong Kong Branch of National Precious Metals Material Engineering Research Centre, Department of Material Science and Engineering, City University of Hong Kong, Hong Kong, China. These authors contributed equally: A. Y. Chen, L. L. Zhu, L. G. Sun. Correspondence and requests for materials should be addressed to A.Y.C. (email: aychen@usst.edu.cn) or to X.Y.W. (email: xianyingwang@usst.edu.cn) or to J.L. (email: jianlu@cityu.edu.hk)

The nanotwin (NT) is strongly thought to be a desirable structure for overcoming the dilemma between strength and ductility[1–4]. However, the strength-ductility synergy of NTed materials is controlled by the deformation mechanisms, involving dislocation pile-up at the twin boundary (TB), partial dislocation slip along the TB, and TB migration, or detwinning[5–10]. Many researchers have explored the coupling mechanisms among these deformation modes[11–16]. For example, Li et al. reported that the plastic deformation transited from inclined dislocation glide to detwinning in NTed Cu when the twin-lamella-spacing ($\lambda$) decreased from 15 to 4 nm[17], and this transition might result in maximum strength in NTed Cu, as observed in experimental tests[18]. Other molecular dynamics (MD) simulations suggested that the movement of partial dislocations, involving 60° dislocation and screw dislocation systems, transited from cutting across TB to dislocation-induced TB migration at a critical $\lambda$ of 2.4 nm[19–21]. Yuan et al.[22] predicted that the Hall-Petch strengthening effect of the secondary twinning deformation was more effective than that of the primary twinning and that there is a critical $\lambda$ of secondary twinning to achieve maximum strength. However, the activation of secondary twinning is still not well understood. It is challenging to clarify the coupling relationship of $\lambda$-dependent deformation behaviors in face-centered cubic (FCC) metals with NTed structure.

As the second generation of advanced high strength steels, metastable austenitic steels such as austenitic stainless steel (SS) and high manganese steel are widely used in the chemical, transportation, and aerospace industries. These steels with NTed structure also exhibit ultrahigh strength, good ductility, and high work hardening[23–28]. Different from the growth NTs in Cu, Ag, and Ni metals and their alloys[19,29,30], the NTs in the structural steels, commonly produced by severe plastic deformation (SPD), are deformation NTs that exhibit a high density of defective structures[31–34]. However, the effects of defective TBs on the deformation behaviors are less understood in nanotwin-strengthened materials. Furthermore, martensitic transformation is the most likely deformation mode in NTed austenitic steels because of the metastable austenite phase[35–39]. Hence, it is difficult to observe or predict the deformation evolution due to indistinguishable structures in postmortem observations or the multiscale diversities in simulations. No direct evidence elucidates how to activate these deformations and what kinetics sequences occur during deformation. Thus, a full-scale deformation map of NTs should be established to provide a clear strengthening mechanism for microstructure manipulation.

In this work, we investigate the $\lambda$-dependent effect on the deformation transitions of NTed steels. Austenitic SS is selected as a template from a structural materials perspective. We performed in situ transmission electron microscopy (TEM) tensile tests to obtain direct evidence of the deformation characteristics, applied a dislocation-based theoretical model and conducted MD simulations to reveal the deformation transition mechanisms of NTs. Both the in situ tests and the simulation results show that there are two critical $\lambda$s to transit the deformation mode from twinning/detwinning to secondary twinning, and then to dislocation accumulation. We generated a deformation map of NTs as a function of $\lambda$, which may aid in the design of advanced steels with superior strength-ductility synergy.

## Results

**Twinning and detwinning by shear stress**. We recorded the deformation modes of the NTs under two types of stress states. In the first, the TBs of the primary NTs exhibit a 70° angle to the loading direction, and a shear stress is applied on the TBs. In the other, the TBs of the primary NTs are closely parallel to the loading direction and endure an axial stress. High density NTs including primary, intersected, and secondary NTs are produced by surface mechanical attrition treatment (SMAT) in type 304 SS. The detailed microstructures and in situ tensile method are given in Supplementary Notes 1 and 2, respectively. As an example, the intersected NTs were selected to demonstrate the typical deformation behaviors of successive twinning and detwinning, as shown in Fig. 1 and Supplementary Movie 1. The intersected bundles of NTs (Fig. 1a) have two systems of TB orientations, where the $TB_1$ exhibits a 70° angle to the loading direction and $TB_2$ is approximately parallel to the loading direction. Supplementary Figure 1 shows the detailed microstructure. In Fig. 1, the red solid lines and arrows indicate the twinning positions and the blue dotted lines and arrows illustrate the detwinning positions. For instance, NTs 1–3 along the $TB_2$ direction and NT 4 along the $TB_1$ direction exhibit partial twinning, as shown in Fig. 1a, b. Similarly, detwinning also occurs, as shown by NTs 2 and 3 (Fig. 1b, c) along the $TB_2$ direction, and NTs 5 (Fig. 1b, c), 6 and 7 (Fig. 1c, d) along the $TB_1$ direction. In addition to these marked NTs, other twinning and detwinning behaviors are indicated in Fig. 1c, d. The statistical results show that coactivation of twinning and detwinning occurs in the intersected NT bundles with $\lambda < 5$ nm under shear stress, and the NTs with $\lambda = 2–3$ nm contribute the highest ratio of 45% (see Supplementary Figure 2).

**Detwinning and martensitic transformation by axial stress**. When the TB is closely parallel to the tensile direction, detwinning of the primary NTs is dominant at first and then martensitic transformation is promoted (see Supplementary Movie 2). Figure 2 displays the in situ TEM observations at a 9° angle between the TB orientation and tensile direction. Figure 2a–f displays the detwinning evolution and Fig. 2e–h indicate the nucleation and growth of the martensite phase. NT 1 is partially dissociated by detwinning, becomes short (Fig. 2b) and vague (Fig. 2c), and finally vanishes (Fig. 2d). NTs 2 and 3 exhibit the same detwinning process, as indicated by the blue arrows in

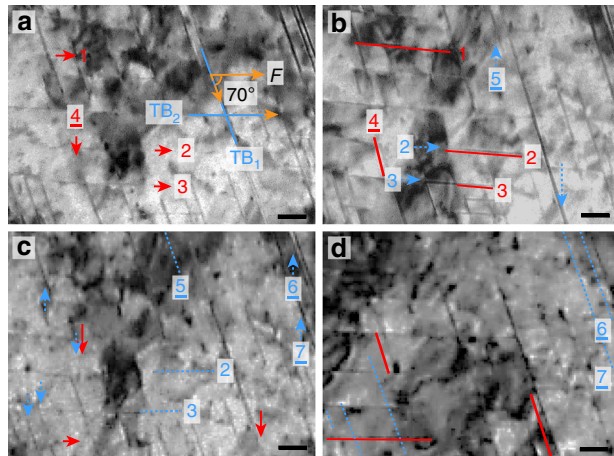

**Fig. 1** Coactivation of twinning and detwinning in the intersected NTs. **a–d** are bright-field TEM images of the intersected NTs, showing the deformation evolution during in situ tensile test. $TB_1$ exhibits a 70° angle to $F$ and $TB_2$ is parallel to $F$, as given in **a**. The red solid lines and arrows, including Lines 1–3 and Arrows 1–3 in the tensile direction and Line 4 and Arrow 4 in the 70° shear direction, indicate the twinning positions. The blue dotted lines and arrows, including Lines 2, 3 and Arrows 2, 3 in the tensile direction and Lines 5–7 and Arrows 5–7 in the 70° shear direction, illustrate the detwinning positions. The standard deviation of $\lambda$ is 0.5 nm. $F$ is the external tensile load. The time frames from **a–d** are 0, 12, 27 and 52 s, respectively. All scale bars are 20 nm

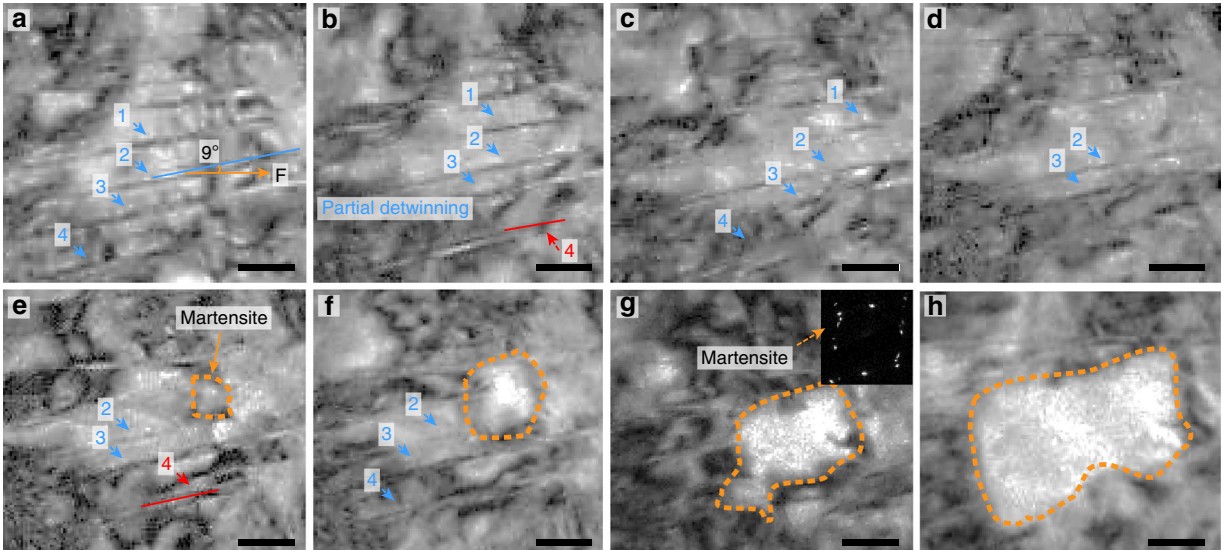

**Fig. 2** Detwinning and martensitic transformation of the NTs. **a** The starting microstructure of the primary NTs with a 9° TB orientation angle to the tensile direction. The blue arrows (Arrows 1–4) illustrate the detwinning positions and the red Line 4 and Arrow 4 indicate the twinning positions in **b** and **e**. **a–f** show the detwinning of NTs 1–4. **e–h** indicate the nucleation and growth of the martensite phase, as marked by orange dotted circles in **e–h**. The inset in **g** is the SAED pattern of the newly-formed martensite phase, as indicated by the orange dotted arrow. The standard deviation of λ is 0.5 nm. F is the external tensile load. The time frames of **a–h** are 0, 10, 17, 22, 37, 42, 63, and 69 s, respectively. All scale bars are 20 nm

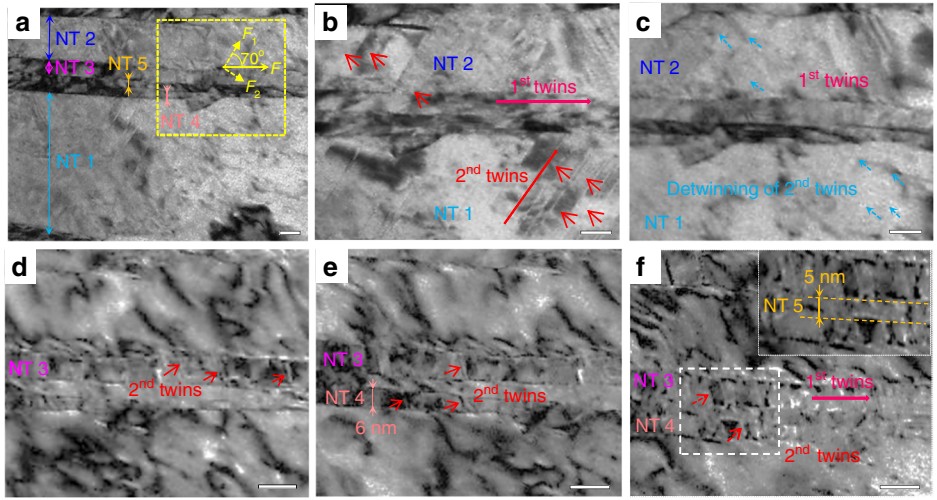

**Fig. 3** Secondary twinning of the NTs. **a** Morphologies of the first NTs 1–5, with λ of 129, 42, 15, 6, 5 nm, respectively. **b–f** Magnifications of the yellow rectangular zone in **a**. Under tensile loading, bundles of the second NTs are initiated in the first NTs 1 and 2 and exhibit detwinning, as shown in **b** and **c**, respectively. After the detwinning of the second NTs in NTs 1 and 2, a new round of secondary twinning and detwinning occurs in NTs 3 and 4, as shown in **d–f**. The inset in **f** is a magnification of the white rectangular zone in **f**, where no secondary twinning was observed in NT 5. F is the external tensile stress, and $F_1$ and $F_2$ are the shear components. The first twins and second twins represent the primary and secondary twins, respectively. The standard deviations of the determined λ are 20 nm for NT 1 and 1 nm for NTs 2–5. The time frames of **a–f** are 0, 5, 7, 10, 15 and 20 s, respectively. All scale bars are 20 nm

Fig. 2b–f. Accompanied by the successive detwinning of NTs 1–3, twinning of NT 4 is observed, as indicated by the red lines and arrows in Fig. 2b, e, respectively. Notably, after the progressive detwinning of NTs 1–3, the martensite phase is nucleated, as indicated by the circle in Fig. 2e. Following the total disappearance of NTs 1–4, the martensite phase starts to grow, as shown in Fig. 2g, h. The selected-area electron diffraction (SAED) pattern in Fig. 2g confirms the newly-formed martensite phase. The detwinning leads to dislocation multiplication by decomposition of the TBs[40], which induces an earlier formation of martensitic transformation than the other coarse grains (CG). The above observations clearly indicate the detwinning of NTs (λ

< 5 nm) preferentially occurs before the martensitic transformation. Thus, the deformation sequence of the NTs (λ < 5 nm) is detwinning and martensitic transformation when the TB is closely parallel to the loading direction.

**Secondary twinning of NTs with λ of 6–129 nm.** When λ increases, secondary twinning inside the primary twins is observed. The primary twin bundles with a TB orientation parallel to the tensile direction are called one-order twins (first twins), where the λ of NTs 1–5 are 129, 42, 15, 6 and 5 nm (Fig. 3a), respectively. The global morphology is given in

Supplementary Figure 3. NTs intersected with first twins at an ~70° angle are secondary twins (second twins), which are much finer than the first twins, as indicated in Fig. 3b. The evolution of secondary twinning is shown in Fig. 3b–f, which are magnifications of the rectangle in Fig. 3a. When loading, the bundles of second twins in NTs 1 and 2 are triggered, as indicated by the red arrows in Fig. 3b. Then, detwinning is progressively activated in NTs 1 and 2, as indicated by the blue arrows in Fig. 3c. Interestingly, NT 3 begins to experience secondary twinning after the detwinning of the second NTs inside NTs 1 and 2, as shown in Fig. 3d. Upon further loading, many second NTs are generated inside NTs 3 and 4 (see Fig. 3e, f), whereas second NTs are difficult to be resolved inside NT 5 under further loading, as shown in the magnification in Fig. 3f. The evolution of secondary twinning is given in Supplementary Movie 3. Notably, (i) the first twins are retained although the loading direction is parallel to the TBs of the first twins, and the formation of second NTs does not annihilate the first twins; (ii) the second NTs exhibit twinning and detwinning inside the first NTs; (iii) the second NTs are activated progressively from the larger NTs 1 and 2 to the smaller NTs 3 and 4; (iv) it is very difficult to induce second NTs in the much finer NT 5 with $\lambda$ close to 5 nm.

**Deformation pattern of multiscale NTs**. The statistical results show that 80% of NTs with $\lambda < 5$ nm (five observed zones) exhibited the coactivation of twinning and detwinning under shear stress, including intersected and primary NTs. 100% of NTs with $\lambda < 5$ nm (two observed zones) exhibited detwinning and the subsequent martensitic transformation when the TBs are closely parallel to the tensile direction. Secondary twinning occurred in primary NTs ($\lambda = 6–129$ nm) with a frequency of 80% among the five observed zones. Only dislocation motion was observed in the left two zones, and the $\lambda$ of these twins are at the submicrometer scale. The statistical diagram is given in Supplementary Figure 4. The above statistical results allowed for generation of a deformation map of the multiscale twins, schematically illustrated in Fig. 4a. When $\lambda$ is smaller than 5 nm, the twinning and detwinning are jointly activated under shear stress and detwinning and martensitic transformation occur successively when the TBs are closely parallel to the loading direction. When $\lambda$ is 6 to 129 nm, secondary twinning is induced inside the primary twin system, and the twinning and detwinning are coactivated only in the secondary twin system. As $\lambda$ increases to the submicrometer scale, dislocation motion is the featured deformation behavior, as observed in TEM tensile tests (see Supplementary Figure 5), where the dislocations emit from the TBs and propagate normal to the loading direction in the form of dislocation walls.

The deformation map of multiscale NTs provides a clear strengthening mechanism for microstructure manipulation. In contrast to the conventional dislocation strengthening in CGs, the twinning, including the secondary twinning, can simultaneously enhance the strength by dislocation pile-up at the TBs and improve the ductility due to partials slipping across TBs[1,18]. The detwinning softens the material by decreasing the TBs, which is similar to the effect of grain growth[6,7]. Thus, the coactivation of twinning and detwinning can sustain successive hardening during plastic deformation[28]. This is a special deformation behavior of NTs on the very fine scale. Accompanied by the detwinning, martensitic transformation is initiated earlier than the other coarse austenite grains, which can further strengthen the materials.

## Discussion

To reveal why there is a confined $\lambda$ range to initiate secondary twinning, we developed a dislocation-based theoretical model and MD simulations, and the results are summarized in Fig. 4b, c.

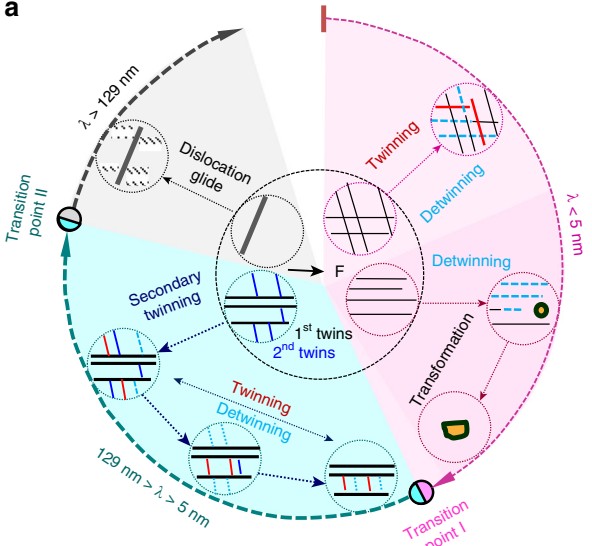

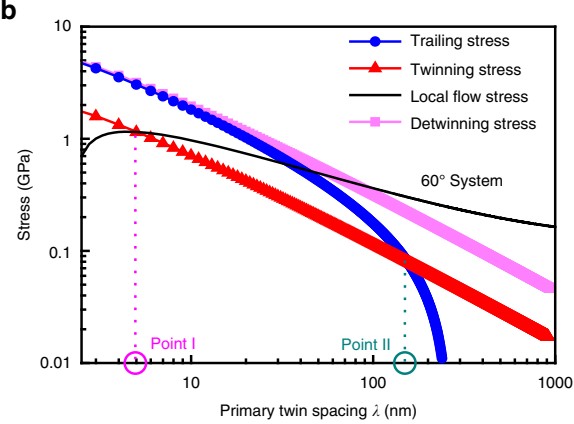

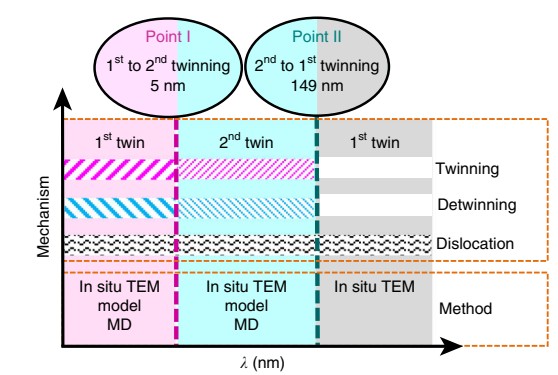

**Fig. 4** Schematic illustration of the deformation modes of NTs. **a** Schematic of the deformation modes observed in our NTed materials by in situ TEM tensile tests. Twinning and detwinning are coactivated under the shear stress, whereas detwinning and martensite transformation present successively under axial stress at the $\lambda$ of the first NTs < 5 nm. Secondary twinning occurs in the first NTs with $\lambda = 6–129$ nm. As $\lambda$ increases, dislocation slip is a dominant deformation behavior. **b** Predicted critical $\lambda$ of the first NTs to generate the second NTs for the 60° partial system using a dislocation-based theoretical model. **c** Deformation modes at each transition zone and proof methods

Figure 4b shows the stress distribution of twinning, trailing, local, and detwinning stresses in a 60° partial system under the condition of the tensile stress parallel to the first TB. According to the theoretical model (see Supplementary Note 3), twinning is

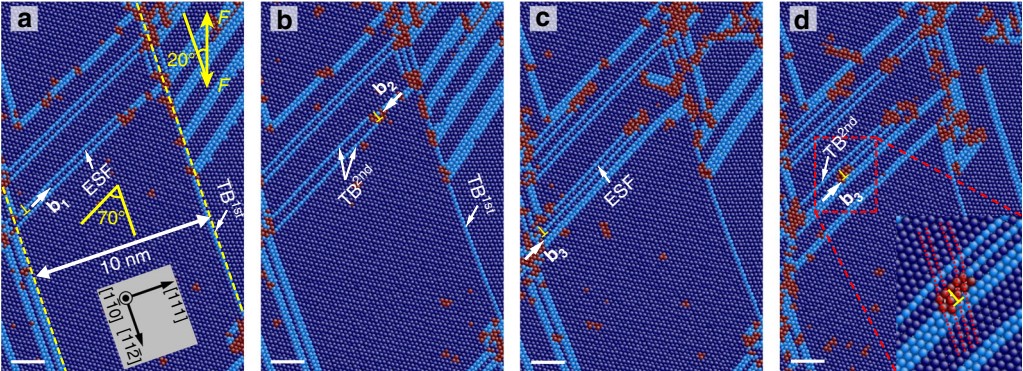

**Fig. 5** Snapshots of secondary twinning in the first NT. **a** Partial dislocation **b1** is ready to propagate along extrinsic stacking fault (ESF). **b** The formation of TBs[2nd] after propagation of **b1** in **a**. **c** Detwinning back to ESF after propagation of **b2** in **b**. **d** The second formation of TB[2nd] owing to the propagation of **b3**. The inset in **d** is an amplified configuration of partial dislocation during propagation. The $\lambda$ of the first NT is 10 nm. $F$ is the uniaxial tensile load. Scale bars are 2 nm

triggered under the condition of twinning stress lower than both the local stress and the trailing stress[5,16]. In the range of 5–149 nm, the twinning stress is lower than both the local stress (upper bound) and the trailing stress (lower bound), thus, the twinning deformation can be initiated. The two critical values, transition points I and II, are very close to the in situ TEM observation results. Furthermore, the theoretical results also predict that the confined range is strongly related to the orientation angle of the TB with the tensile stress and the types of partial dislocations. For example, the $\lambda$ is 4.8–186 nm for the 60° system at 85.2°, and 6.7–237 nm for the screw system at 26.5° (see Supplementary Figure 6), where the two orientation angles are thought to easily trigger NTs in the two partial systems[15,16]. Notably, the twinning stress decreases linearly with the increase of $\lambda$, suggesting that the NTs with larger $\lambda$ generate secondary twins more easily. That is why the secondary twinning is initiated from NTs with a larger $\lambda$ to those with a smaller $\lambda$, as shown in Fig. 3.

A series of MD simulations ($\lambda = 2.5$, 3.74, 5, 10, 15 nm) was carried out to investigate the deformation transition mechanisms (see Supplementary Note 4). Interestingly, only twinning and detwinning of the first NTs were observed in the $\lambda = 2.5$, 3.74 nm samples whereas twinning and detwinning of the second NTs was triggered in the $\lambda = 10$, 15 nm samples. The $\lambda = 10$ nm sample was selected as a representative to illustrate the twinning and detwinning processes of the second NTs (Fig. 5). The angle between first and second NTs is 70°, which agrees with our experimental observations. The twinning partial **b$_1$** first propagates from the left side along the extrinsic stacking fault (ESF) (Fig. 5a). Secondary twinning occurs, accompanied by the propagation of **b$_1$** to the right side (Fig. 5b). In the next stage, another partial **b$_2$** propagates along the newly-formed TB[2nd] from the right to the left side (Fig. 5b), leading to detwinning back to ESF (Fig. 5c). Thereafter, the propagation of a partial **b$_3$** contributes to the formation of TB[2nd] (Fig. 5d). The amplified red dashed region clearly shows the configuration of the partial **b$_3$** during propagation. For comparison, a snapshot of the relaxed configurations before deformation is provided in Supplementary Figure 7. More interestingly, the MD simulation results show that twinning/detwinning of the first and second twins both occurred in the $\lambda = 5$ nm sample (see Supplementary Figure 8 and 9). This result is very close to the lower limit of $\lambda$ in the in situ observations.

The intrinsic mechanisms to trigger twinning/detwinning were elucidated by considering two active partials of 60° and screw systems. It is difficult to separate and probe the 60° dislocation and screw dislocation with in situ TEM tests due to the high

density of preset dislocations[33], but they are the two classic types of dislocation systems to activate the twinning/detwinning deformation[5,16]. Furthermore, the activation of twinning/detwinning depends on the magnitude of twinning, trailing, detwinning and local stress and is influenced by the orientation angle of the TBs with loading direction. Figure 6 shows the stress distribution of the two partial systems with orientation angles of 70° (a, b) and 9° (c, d), as observed in TEM observations. The NT with $\lambda = 2.5$ nm was selected as a typical scale and the zone of unit length ($L$) was 100–500 nm according to the visual sight of the TEM observations. When the twinning stress is smaller than both the trailing stress and the local stress, twinning is initiated. Similarly, when the detwinning stress is less than the local stress, detwinning occurs. In Fig. 6a, the twinning stress is smaller than the trailing stress at $L < 160$ nm and larger than the trailing stress in most of the observed area ($L > 160$ nm), suggesting that twinning of the 60° partial system occurs with a low probability. However, the detwinning stress is lower than the local stress in the observed area ($L > 40$ nm), thus, detwinning is dominant. In contrast to the 60° partial system, the trailing stress of the screw system is negative at $L < 370$ nm and larger than the twinning stress at $L > 370$ nm, therefore, twinning occurs in the entire observed area (Fig. 6b). Correspondingly, the detwinning stress is also negative in the entire area, therefore, detwinning is impossible. The above results show that the 60° system is susceptible to detwinning (Fig. 6a) and twinning easily occurs in the screw system (Fig. 6b), which can explain the coactivated twinning and detwinning deformation under shear stress at a 70° angle of the TB orientation to loading direction. The MD results of the $\lambda = 2.5$ nm sample also demonstrate the coactivation of twinning and detwinning behaviors, as shown in Supplementary Figure 10 and Supplementary Movie 4.

In the case of a 9° angle, twinning is very hard to initiate for the 60° system due to a larger twinning stress than the trailing stress (Fig. 6c), and detwinning is the most possible deformation mode due to the smaller detwinning stress. However, for the screw system in Fig. 6d, twinning is only activated at $18$ nm $< L <$ $180$ nm. This means that twinning has a low probability since twinning is prohibited in most of the observed area ($180$ nm $< L$ $< 500$ nm). However, detwinning of the existing NTs is triggered in the entire observed area ($L > 60$ nm). Thus, for the screw system at a 9° angle of the TB to the loading direction, detwinning is the dominant deformation behavior in the observed area, and twinning is also activated at a low frequency. Finally, the NTs disappear due to the overwhelming majority of detwinning compared to twinning.

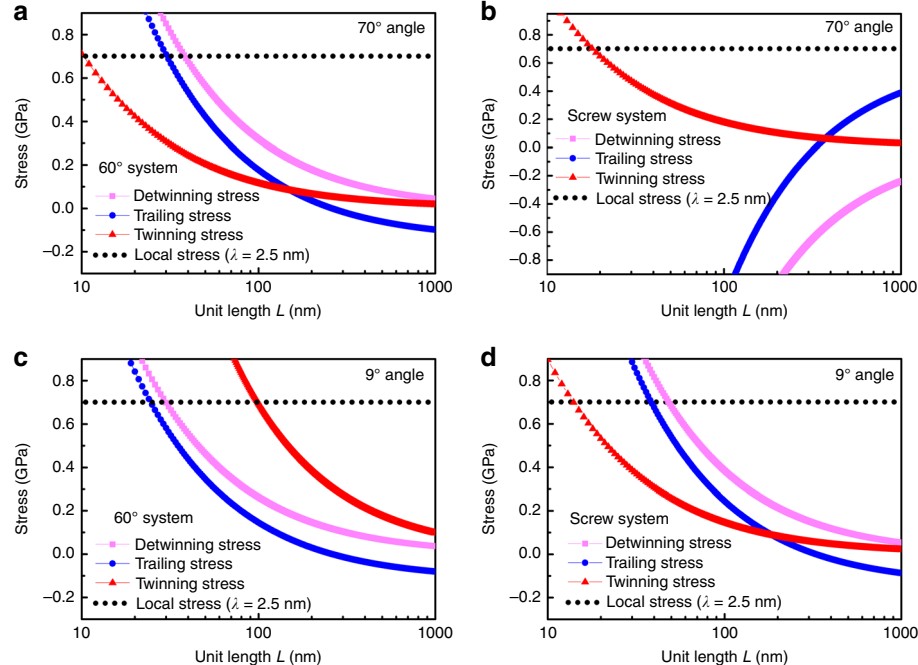

**Fig. 6** Distribution of twinning and detwinning stresses during tensile deformation. 70° (**a**, **b**) and 9° (**c**, **d**) angles of the first TB to the tensile direction. **a**, **c** 60° system. **b**, **d** Screw system. The activation of twinning requires the twinning stress lower than both the local stress and trailing stress, thus, twinning is prohibited at $L > 180$ nm in **a**. When the detwinning stress is smaller than the local stress, detwinning occurs. Therefore, detwinning is dominant in **a** ($L > 40$ nm). On the opposite, twinning in **b** is the featured deformation behavior due to the negative trailing and detwinning stresses in the screw system. Detwinning controls the deformation in **c** ($L > 40$ nm) and **d** ($L > 60$ nm), whereas twinning occurs in **d** in a smaller range ($L < 180$ nm)

To conclude, we established a deformation map of NTs with a $\lambda$ from several nanometers to hundreds of nanometers using in situ TEM observations combined with a theoretical model and MD simulations. For NTs with $\lambda < 5$ nm, twinning and detwinning are concurrently activated under shear stress. The partial dislocation of the 60° system is primarily responsible for the detwinning, whereas the screw system exhibits twinning deformation. Detwinning is dominant when the TB orientation is closely parallel to the tensile direction. The formation of secondary twinning has a critical range of $\lambda = 6$–129 nm, which is highly consistent with the results of the dislocation-based theoretical model and MD simulations. Our methods are not specific to particular metals, so the conclusion should be generally applicable to other NTed FCC metals.

## Methods

**Specimen preparation and in situ tensile tests**. The NTed 304 SS was produced by SMAT with the GCr15 balls randomly impacted onto the specimen surface with an impacting frequency of 20 kHz. The detailed processing parameters and microstructure features are given in ref. [33]. The specimens for in situ tests were taken 150 μm from the surface and prepared by mechanical polishing to 30 μm thickness. The 5 mm × 2 mm × 30 μm foils were thinned by twin-jet electro-polishing, as illustrated in Supplementary Figure 11. In situ TEM tensile tests were performed in a JEM-2100 TEM operated at 200 kV using a Gatan 654 straining holder under manual control. The applied load, $F$, was uniaxial along the horizontal direction. Four samples with 14 microzones were observed through in situ TEM experiments, where seven observed zones had NTs with $\lambda < 5$ nm, five observed zones had NTs with 5 nm $< \lambda < 129$ nm, and two observed zones contained twins with $\lambda$ at the submicrometer scale. Details of the sample preparation, dimension and movies are given in Supplementary Note 2.

**Dislocation-based theoretical model and MD simulations**. A dislocation-based theoretical model (see Supplementary Note 3) was proposed to explain the twinning, detwinning, and secondary twinning behaviors through comparing the twinning stress with the trailing, detwinning, and local stresses. MD simulations (see Supplementary Note 4) were carried out using the large-scale atomic/molecular massively parallel simulator (LAMMPS) code[41]. An embedded atom method potential for the Fe–Ni–Cr system was employed to describe the atomic interaction

in the samples[42,43]. The MD simulation results exhibited in this work were colored complying with the following standard: dark blue, light blue, and red represent FCC, hexagonal close packed and disordered atoms, respectively. The models were prepared to be an FCC structure with a lattice constant of 3.562 Å. The relaxed samples were subjected to uniaxial tensile simulation along the y-direction with 0° and 70° angles between the TB orientation and tensile direction.

## Data availability

The authors declare that all data generated or analyzed during this study are included in this published article (and its Supplementary Information files), and the data are available from the corresponding author on reasonable request.

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

## Acknowledgements

We gratefully acknowledge the supports from the National Key R&D Program of China (Project No. 2017YFA0204403), and the National Natural Science Foundation of China (grants 51771121, 51590892, 11472243, 11621062, and 51572173). Prof. Wang acknowledges the funding provided by the Shanghai Municipal Science and Technology Commission (16060502300, 16JC402200, and 18511110600). L.L.Z, H.T.W., and J.B.L. thank the Fundamental Research Funds for the Central Universities (2018XZZX001–05).

## Author contributions

J.B.L. and A.Y.C. carried out the in situ TEM experiments. L.G.S. performed the molecular dynamics simulations. L.L.Z. developed the dislocation-based theoretical model and analyzed the deformation mechanisms. H.T.W., J.H.Y., and X.Y.W. conceived and designed the experiments and simulations. J.L. contributed to the data analysis and experimental design. A.Y.C., L.L.Z., and L.G.S. wrote the manuscript with contributions from the other authors. All authors commented on the final manuscript and conclusions of this work.

## Additional information

**Competing interests:** The authors declare no competing interests.

**Journal Peer Review Information**: *Nature Communications* thanks the anonymous reviewers for their contribution to the peer review of this work. Peer reviewer reports are available.

