## [Peer review file · Nature Communications]

Reviewers' comments:

Reviewer #1 (Remarks to the Author):

The authors revealed the relationship between the twin size and the deformation mechanism through in-situ TEM experiments, theoretical analysis and MD simulations and built up a fairly comprehensive deformation pattern in the stainless steel. These results in the manuscript may serve as a significant guidance for understanding the deformation behaviors in nano-twinned metals and designing advanced structural materials with excellent mechanical performances. Thereby, in my own opinion, this manuscript can be accepted after some minor revisions listed below.

1. The in-situ TEM image showed in this manuscript were highly consistent with the simulation results. Could the authors provide ballpark fractions about nano-twins with different sizes corresponding to different deformation mechanisms in the whole observed area? I guess the statistic results might be more solid to support the conclusion rather than the single example.
2. The nucleation of martensite after the detwinning mentioned in the results. Could the authors give some discussion about the internal connection between them? It might also be a good topic to explore in the future work.
3. The detwinning might lead to the softening in metals. Could the authors add some discussion about the strength-softening effects from different deformation mechanisms?
4. There are some errors in the reference list like the authors in the reference [2], please re-check that.

Reviewer #2 (Remarks to the Author):

The study of nanotwin-related deformation behaviors is very important to understand the mechanical properties of nanotwinned metals and to potentially evade the dilemma between strength and ductility. As pointed out by the authors, many nanotwin-related deformation mechanisms including twinning partial dislocations, TB migration, detwinning and secondary twinning, have been widely investigated in recent years using a combination of experiments and simulations. This paper is focused on understanding the dependence of deformation behavior of NTed steel on twin-lamella-spacing (λ). By combining in-situ TEM tensile tests, a dislocation-based theoretical model and MD simulations, the authors reveal the lower and upper λ limits for the formation of secondary twinning in the NTed steel. These observations will enrich the understanding of mechanical behaviors of NTed metals. However, concerning the novelty, the mechanism transition (dislocation slip versus twinning/detwinning) presented in this paper does not represent significant advance to the field in comparison to previous works by the same group on the same material (304 stainless steel).

Moreover, essential microstructure characterization of the NTed SS is incomplete. Is there a texture in the SMAT specimens or not? Is the microstructure stable? What is the dislocation density before and after test? What is the loading direction to the nanotwins? How are the in-situ foils cut from the NTed SS specimen? The authors did not provide details of the texture and microstructure evolution before and after loading. Statistical analysis of twin/slip system activation, twin-lamella-spacing distribution or dislocation density estimation, which are important to understand the mechanism transition, are not provided. Some additional crystal plasticity simulations may be helpful to understand the transition in deformation mechanisms in NTed steel because in-situ experiments may be challenging to provide the entire picture.

The authors show that detwinning of the primary twins is dominant when external stress is closely parallel to the TBs. This is not reasonable because there is no shear stress to drive the movement of partial dislocations. In-situ tests show that detwinning is followed by martensitic transformation, but is missing in the modeling.

What is the lattice constant chosen for the MD simulation? Are the simulation results sensitive to the EAM potential used in the presented MD simulations which was designed for FeNiCr instead of SS? TBs could not be identified in Fig. S10. How did the authors identify twinning and detwinning partial dislocations in the simulations? Additional videos showing detwinning and twinning processes are strongly suggested.

Theoretical model and MD simulations are considered to be essential to the understanding of the

deformation mapping presented in this work. However, most discussion has been buried in the supplementary information, which makes the related sections difficult to follow. The paper has other presentation issues. All figure captions in the main text are extremely oversimplified and should contain sufficient information so that the reader can understand the figure without frequently referring to the main text. The writing of the manuscript is careless (e.g., page 8 line 150).

To this referee, this work does not provide necessary microscopy information, statistical analysis, and computational data to support the conclusion. Therefore, it is not suitable for publication in Nature Communications.

Reviewer #3 (Remarks to the Author):

In this work, the authors presented a deformation mechanism map of nanotwinned austenitic stainless steel using in-situ TEM tensile tests, dislocation-based theoretical modeling, and MD simulations. With twin spacing as the controlling parameter, the deformation map shows that the FCC stainless steel deforms through co-activated twinning/detwinning at twin spacing less than 5 nm, secondary twinning at twin spacing 5 – 129 nm, and dislocation glide at twin spacing larger than 129 nm. While this map is important to understand the deformation mechanism of nanotwinned materials, the authors failed to prove that this map is indeed what they claimed to be due to the many issues discussed in detail below.

1. My major concern is related to the twin spacing measurement. In Figures 1, 2, and S1, it seems to me that the authors only measured the twin spacing of the very thin twin laminates to make the sample lies in the region of less than 5 nm. In fact, the large white areas in Figures 1, 2, S1 (a) indicate that much larger twins exist with a twin spacing of about 20 nm based on my rough measurement. In addition, the authors also only measured the white line in the dark-field TEM shown Figure S1 (c), without considering the large dark areas. If my understanding is correct, this sample cannot be considered as having only twin spacing less than 5 nm. In contrast, the authors measured all the areas in the sample containing twin spacing 5-129 nm, as shown in Figure 3 (a). It sounds to me that the authors were trying to divide the two samples into different categories by using different measuring methods.

2. When discussing the deformation mechanism for twin spacing between 5 and 129 nm, the authors only used one sample containing a lot of twins, with twin spacing varying from 5 to 129 nm. However, the authors meant to conclude that the deformation mechanism (secondary twinning) happened in this sample will happen in the nanotwinned samples as long as their twin spacing is between 5 and 129 nm. In other words, a sample containing only 6 or 7 (or any number between 5 and 129) nm twins will deformation through secondary twinning. This can be evidenced by their dislocation-based theoretical modeling and the MD simulations, where only one twin spacing is included each time. The authors need to provide more experimental evidence to support their conclusion by doing in-situ TEM tensile tests on more samples containing different twin spacing.

3. Only 60-degree and screw dislocations were considered in the dislocation-based theoretical model to explain the effect of twin spacing on the twinning, detwinning, etc stress and thus on the change of deformation pattern. But the authors did not mention anything that these two kinds of dislocations are involved in the TEM tests. The theoretical modeling told nothing if such kinds of dislocations did not happen. It is of course highly possible that these dislocations are involved, but the authors at least should say something to make the connection between theoretical modeling and TEM tests. In addition, it should not be just the 60-degree dislocation and screw dislocations that are responsible for the twinning, detwinning etc. Other dislocation types such as edge dislocation, mixed dislocation should also be considered in the theoretical modeling. If by any chance there are technical difficulties dealing with these dislocations, the authors should at least discuss the situation in their modeling work.

4. In the dislocation-based modeling, the authors defined the 60-degree dislocation as the case when $\alpha_1=90$ and $\alpha_2=30$. That way the angles between the Burgers vectors of the leading and trailing partial would be $90+30=120$, which seems against my understanding. For fcc metals, the angle between the two Burgers vectors should always be 60 degrees, independent of the dislocation type.

5. The TEM images in Figure S3 is not clear. Although the authors claim this is an image for a sample containing twins in submicrometer scale, it seems to me that this image contains a twin with a spacing of about 100 nm, less than 129 nm.

6. In Page 11, lines 208-212, the authors claimed that detwinning is the dominant deformation

mechanism for the screw dislocation. This is however inconsistent with Figure 6d showing that detwinning stress is highest in the whole zone and thus detwinning is least likely.

7. What is the crystal orientation perpendicular to the paper for the TEM images? Without this orientation, the angle between the applied force and the TB orientation is not well-defined and it is not appropriate to use the theoretical model and MD simulations to explain experimental results.

8. Exactly what did the authors mean by "external stress," "horizontal stress," "external stress" etc? Are they tensile stress or shear stress.

9. What did the authors mean by "incident twin boundary (ITB)" in Figure S1? And which TB is CTB, which TB is ITB?

10. What does the red shaded area mean in Figure 6?

11. Page 10, lines 186, the authors claimed that the twinning/detwinning in this work is very special. What did they mean by "very special?"

12. Page 10, lines 193–201. When the authors discussed the zone effect on twinning/detwinning, why the authors gave an upper or lower bound in the discussed zone. For example, that the trailing stress is smaller than the twinning stress at 160-500 nm length, is also true for the length larger than 500 nm. Then why the authors limited the zone to 160–500 nm, instead of just >160 nm. Similarly, 100-160 nm can also be <160 nm, 100-370 nm can be <370 nm, etc.

13. In MD simulations, what do the sample configuration and twin structure looks like before deformation? The authors only gave the snapshot after deformation, which is clearly not enough to help understand the simulation results.

14. Page 11, lines 222-223. The authors claimed that their conclusion should be generally applicable to other NTed fcc metals. This is actually inconsistent with their claim on Page 10, line 186, that their "successive twinning/detwinning found in this work is very special."

15. The title is confusing. I don't understand why the authors used the word "holographically" in their title. It seems to me that the authors were trying to say that their map contains more information than the usual map probably mentioned in the literature work. Even this is the case, the word is still confusing and ambiguous.

16. Page 5, line 88, what does "one-system" mean here?

17. Page 6, line 112, why there is 5 nm twin in nanotwins with spacing in the range of 6-129 nm?

18. Page 7, line 136, is "dislocation motivation" actually "dislocation activation?" The authors used "dislocation motivation" in a lot of places in the main text and supplementary material.

19. Page 12, line 236. A reference is needed for the LAMMPS code and the EAM potential for Fe-Ni-Cr.

20. There are a lot of grammar issues in this manuscript. For example, Page 3, line 40, "dependence" should be "dependent;" Page 8, line 144, "predict" should be "predicted;" Page 10, line 188, "elucidate" should be "elucidated" etc.

Overall, this manuscript is not well-prepared and, as a result, the story is not well-presented. I can get what the authors were trying to prove but I am confused about so many details. On one hand, the authors should provide more evidence to prove their theory; on the other hand, they should also try to draft this paper in a clear and lucid style.

Response to the Reviewers' Comments

Ref. No.: NCOMMS-18-14200

Title: Mapping the deformation behaviors of nanotwins: scale law of complex deformation transitions

Authors: A. Y. Chen, L. L. Zhu, L. G. Sun, J. B. Liu, H. T. Wang, X. Y. Wang, J. H. Yang, J. Lu

Reviewer #1 (Remarks to the Author):

Reviewer's comment: The authors revealed the relationship between the twin size and the deformation mechanism through in-situ TEM experiments, theoretical analysis and MD simulations and built up a fairly comprehensive deformation pattern in the stainless steel. These results in the manuscript may serve as a significant guidance for understanding the deformation behaviors in nano-twinned metals and designing advanced structural materials with excellent mechanical performances. Thereby, in my own opinion, this manuscript can be accepted after some minor revisions listed below.

Reviewer's comment: 1.1. The in-situ TEM image showed in this manuscript were highly consistent with the simulation results. Could the authors provide ballpark fractions about nano-twins with different sizes corresponding to different deformation mechanisms in the whole observed area? I guess the statistic results might be more solid to support the conclusion rather than the single example.

Authors' answer: That is a very valuable suggestion to give a statistic analysis of deformation features. Factually, we carried out four samples in *in-situ* TEM tests, and 14 micro-zones were observed. The results we gave in the manuscript were the dominant observations. The corresponding descriptions of the samples were added in

Page 14 Line 3: “Four samples with 14 micro-zones were observed through *in-situ* TEM experiments, where 7 observed zones had the NTs with $\lambda < 5$ nm, 5 observed zones were the NTs with $5 \text{ nm} < \lambda < 129$ nm, and 2 observed zones were the twins with λ at submicrometer scale.”.

The detailed statistic results of the deformation features are given in **Page 5 Line 8:**

“The statistic results show that co-activation of twinning and detwinning occurs in the intersected and primary NT bundles with λ smaller than 5 nm (with evaluated error of 0.5 nm) under shear stress, and the NTs with $\lambda=2-3$ nm contribute the highest ratio of 45% (see Supplementary Fig. S2).” and **Page 7 Line 17:** “The statistic results exhibit

that 80% NTs with $\lambda < 5$ nm (5 observed zones) exhibit the co-activation of twinning and detwinning under shear stress, including intersected and primary NTs. While 100% NTs with $\lambda < 5$ nm (2 observed zones) exhibit detwinning and the subsequent martensitic transformation when the TBs are closely parallel to the tensile direction.

Secondary twinning occurs in primary NTs ($\lambda=6-129$ nm) with a frequency of 80% among the 5 observed zones. Only dislocation motion is observed in the left 2 observed zones, where the λ of those twins are at submicrometer scale. The statistic diagram is given in Supplementary Fig. S4.”.

The added supplementary figures (Supplementary Fig. S2 and S4.) are given below:

Supplementary Figure S2. Statistic distribution of λ characterized by co-activation of twinning and detwinning in the intersected NTs. The NTs with $\lambda < 5$ nm are *in-situ* observed co-activation of twinning and detwinning, and the NTs with $\lambda=2-3$ nm contribute the highest ratio. The statistical errors of the evaluated λ are 0.5 nm.

Supplementary Figure S4. The statistic diagram of the deformation behaviors.

14 micro-zones are observed through *in-situ* TEM experiments, where 7 observed zones have the NTs with $\lambda < 5$ nm, 5 observed zones are the NTs with $5 \text{ nm} < \lambda < 129$ nm, and 2 observed zones are the twins with λ in ultrafine scale, as given in a. The determined errors of λ are 20 nm for upper limit of 129 nm and 1 nm for the lower limit. The statistic results exhibit that 80% NTs (5 observed zones with $\lambda < 5$ nm) are co-activated twinning and detwinning under 70° angle of TB to loading direction. 100% NTs (2 observed zones with $\lambda < 5$ nm) exhibit detwinning and the subsequent martensite transformation under 9° angle of TB to loading direction. Among the 5 observed zones containing NTs with $\lambda = 5$ -129 nm, secondary twinning occurs with a frequency of 80%. In the last 2 observed zones, dislocation motion is active in the NTs with λ in ultrafine scale.

Reviewer's comment: *1.2. The nucleation of martensite after the detwinning mentioned in the results. Could the authors give some discussion about the internal connection between them? It might also be a good topic to explore in the future work.*

Authors' answer: We added some explanations about the formation of martensitic transformation, as given in **Page 6 Line 7**: “The detwinning leads to dislocation multiplication by the decomposition of TBs⁴⁰, which induces the early formation of martensitic transformation compared with the other coarse grains (CG).” and in **Page 9 Line 1**: “Accompanied with the detwinning, martensitic transformation is initiated

earlier than the other coarse austenite grains, which can further strengthen the materials.”.

Reviewer's comment: *1.3. The detwinning might lead to the softening in metals. Could the authors add some discussion about the strength-softening effects from different deformation mechanisms?*

Authors' answer: As the reviewer’s suggestion, we added some discussions about the strengthening-softening effects of different deformation mechanisms, as given in **Page 8 Line 15:** “Different from the conventional dislocation strengthening in CGs, the twinning, including the secondary twinning, can simultaneously enhance the strength by the dislocation pile-up at TBs and improve the ductility by partials slipping cross TBs^{1,18}. While the detwinning softens the material by the decrease of TBs, which is similar to the effect of grain growth^{6,7}. Thus, the co-activation of twinning and detwinning can sustain successively work hardening and plastic deformation without failure²⁸. That is the special deformation behavior of NTs in the very fine scale. Accompanied with the detwinning, martensitic transformation is initiated earlier than the other coarse austenite grains, which can further strengthen the materials.”.

Reviewer's comment: *1.4. There are some errors in the reference list like the authors in the reference [2], please re-check that.*

Authors' answer: As kindly mentioned by the reviewer, we corrected the reference [2], as shown below:

The reference [2] is changed from “2. Sansoz, F., Lu, K., **Zhu, Y. T.**, & Misra, A. Strengthening and plasticity in nanotwinned metals. *MRS Bulletin* **41**, 292-297 (2016).” into “2. Sansoz, F., Lu, K., **Zhu, T.**, & Misra, A. Strengthening and plasticity in nanotwinned metals. *MRS Bulletin* **41**, 292-297 (2016).”.

Reviewer #2 (Remarks to the Author):

Reviewer's comment: *2.1. The study of nanotwin-related deformation behaviors is very important to understand the mechanical properties of nanotwinned metals and to potentially evade the dilemma between strength and ductility. As pointed out by the authors, many nanotwin-related deformation mechanisms including twinning partial dislocations, TB migration, detwinning and secondary twinning, have been widely investigated in recent years using a combination of experiments and simulations. This paper is focused on understanding the dependence of deformation behavior of NTed steel on twin-lamella-spacing (λ). By combining in-situ TEM tensile tests, a dislocation-based theoretical model and MD simulations, the authors reveal the lower and upper λ limits for the formation of secondary twinning in the NTed steel. These observations will enrich the understanding of mechanical behaviors of NTed metals. However, concerning the novelty, the mechanism transition (dislocation slip versus twinning/detwinning) presented in this paper does not represent significant advance to the field in comparison to previous works by the same group on the same material (304 stainless steel). Moreover, essential microstructure characterization of the NTed SS is incomplete.*

Authors' answer: In our previous works, we had studied how to obtain the nanotwinned structure in 304 stainless steel (SS) and explored the strength-ductility relationship of the NTed 304 SS. The essential microstructure characterization of the NTed SS had been reported in our previous work in Ref. 33 (Chen, A. Y., Ruan, H. H., Wang, J., Chan, H. L., Wang, Q., Li, Q., & Lu, J. The influence of strain rate on the microstructure transition of 304 stainless steel. *Acta Mater.* **59**, 3697-3709 (2011).), where we reported that relationship of microstructure with impact velocity of the balls by surface mechanical attrition treatment (SMAT), and calculated the critical impact stress to form the nanotwins. Then we investigated the mechanical properties of the NTed 304 SS with different gradient structure, and found that these SMATed samples with different NTed structure exhibited ultra-high strength, good ductility and high

work hardening (Ref. 28, Chen, A. Y., Liu, J. B., Wang, H. T., Lu, J., & Wang, Y. M. Gradient twinned 304 stainless steels for high strength and high ductility. *Mater. Sci. Eng. A* **667**, 179-188 (2016). On the basis of these works, we found that the NTs had multiscale (from several nanometers to hundreds of nanometers) and multiform (primary, secondary, and hierarchical) features, thus, the deformation behaviors of these deformation twins exhibited much more complicated. In order to provide a clear strengthening mechanism for the microstructure manipulation, a full-scale deformation map of the NTs with size from several nanometers to hundreds of nanometers should be established. In this work, we explore the transition mechanisms of the deformation nanotwins on the basis of *in-situ* TEM observations combined with theoretical model and MD simulations, and construct a deformation map according to the direct evidences and the simulation results.

Reviewer's comment: 2.2. *Is there a texture in the SMAT specimens or not?*

Authors' answer: There is no texture in the SMATed specimens. The balls randomly impacted on the specimen surface during SMAT, therefore, no texture was newly formed in the SMATed sample. Our previous works (Ref. 33) also evidenced that no texture was formed in the specimens by XRD analysis, SEM and TEM observations. The corresponding explanations are given in **Page 13 Line 16**: “The NTed 304 SS was produced by SMAT, where the GCr15 balls randomly impacted onto the specimen surface with an impacting frequency of 20 kHz. The detailed processing parameters and microstructure features were given in Ref³³”.

Reviewer's comment: 2.3. *Is the microstructure stable?*

Authors' answer: The nanotwins in 304 SS are stable. The nanotwins in 304 SS were thermally stable up to 650 °C, as reported in our previous work (A.Y. Chen, W.F. Hu, et al., Improving the intergranular corrosion resistance of austenitic stainless steel by high density twinned structure, *Scripta Materialia* 130 (2017) 264-268). The other research work found that the deformation nanotwins could endure the heat treatment high up to 800 °C (F.K. Yan, G.Z. Liu, N.R. Tao, K. Lu, *Acta Mater.* 60 (2012) 1059).

Reviewer's comment: 2.4. *What is the dislocation density before and after test?*

Authors' answer: Before test, the dislocation density of the nanotwinned sample is high up to $2 \times 10^{16} \text{ m}^{-2}$ estimated from HRTEM images, as reported in our previous work (Ref. 33, Chen, A. Y., Ruan, H. H., Wang, J., Chan, H. L., Wang, Q., Li, Q., & Lu, J. The influence of strain rate on the microstructure transition of 304 stainless steel. *Acta Mater.* **59**, 3697-3709 (2011)). Generally, the dislocation density should increase after tensile deformation. However, the dislocation density of the deformation nanotwins after *in-situ* tensile was hard to be calculated because the starting material already had very high dislocation density.

Reviewer's comment: 2.5. *What is the loading direction to the nanotwins?*

Authors' answer: In *in-situ* tests, the loading was applied along the horizontal direction macroscopically. Two types of stress states applied to the nanotwins are recorded. One is that the twin boundaries exhibit a 70° angle to the loading direction (Fig. 1), then co-activation of twinning and detwinning is found. The other is that the twin boundaries are closely parallel to loading direction (Fig. 2), then detwinning and martensitic transformation are observed.

These explanations are added in **Page 4 Line 11**: “We record the deformation modes of the NTs under two types of stress states. One is that TBs of the primary NTs exhibit a 70° angle to the loading direction, thereafter, a shear stress is applied on the TBs. The other is that the TBs of the primary NTs are closely parallel to the loading direction, and endure an axial stress.” and in **Page 14 Line 2**: “The applied load is uniaxial along the horizontal direction.”.

Reviewer's comment: 2.6. *How are the in-situ foils cut from the NTed SS specimen?*

Authors' answer: The samples for *in-situ* tests were prepared by mechanical polishing to 150 μm depth from the surface, and then thinned also by mechanical

polishing from the other side to a final thickness of 30 μm . The foils with a dimension of 5 mm \times 2 mm \times 30 μm were thinned by twin-jet electropolishing in ethanol solution containing 5% perchloric acid at -30 $^{\circ}\text{C}$. The added descriptions are given in **Page 13 Line 18**: “The specimens for *in-situ* tests were taken out at 150 μm depth from the surface, and prepared by mechanical polishing to 30 μm thickness. Then foils with a dimension of 5 mm \times 2 mm \times 30 μm were thinned by twin-jet electropolishing, as illustrated in Supplementary Fig. S11.” and in the supplementary information **Page 13 line 5**: “The samples for *in-situ* tests were prepared by mechanical polishing to 150 μm depth from the surface, and then thinned also by mechanical polishing from the other side to a final thickness of 30 μm . The foils with a dimension of 5 mm \times 2 mm \times 30 μm were thinned by twin-jet electropolishing in ethanol solution containing 5% perchloric acid at -30 $^{\circ}\text{C}$, as shown in Supplementary Fig. S11.”.

Reviewer's comment: 2.7. *The authors did not provide details of the texture and microstructure evolution before and after loading.*

Authors' answer: As we explain above (**Responses 2.1 and 2.2**), there is no texture existing in the specimens. The microstructure before and after *in-situ* tests are given in the *in-situ* TEM images, and the videos.

Reviewer's comment: 2.8. *Statistical analysis of twin/slip system activation, twin-lamella-spacing distribution or dislocation density estimation, which are important to understand the mechanism transition, are not provided.*

Authors' answer: In this work, the suggested deformation modes are statistic results, as explained above (**Response 1.1**), and added in **Page 5 Line 8**: “The statistic results show that co-activation of twinning and detwinning occurs in the intersected and primary NT bundles with λ smaller than 5 nm (with evaluated error of 0.5 nm) under shear stress, and the NTs with $\lambda=2-3$ nm contribute the highest ratio of 45% (see Supplementary Fig. S2).” and **Page 7 Line 17**: “The statistic results exhibit that 80% NTs with $\lambda < 5$ nm (5 observed zones) exhibit the co-activation of twinning and

detwinning under shear stress, including intersected and primary NTs. While 100% NTs with $\lambda < 5$ nm (2 observed zones) exhibit detwinning and the subsequent martensitic transformation when the TBs are closely parallel to the tensile direction. Secondary twinning occurs in primary NTs ($\lambda=6-129$ nm) with a frequency of 80% among the 5 observed zones. Only dislocation motion is observed in the left 2 observed zones, where the λ of those twins are at submicrometer scale. The statistic diagram is given in Supplementary Fig. S4.”.

The detailed microstructures of the NTed specimen, including twin-lamella-spacing distribution and dislocation density, have been reported in our previous papers (Ref. 33, Chen, A. Y., Ruan, H. H., Wang, J., Chan, H. L., Wang, Q., Li, Q., & Lu, J. The influence of strain rate on the microstructure transition of 304 stainless steel. *Acta Mater.* **59**, 3697-3709 (2011).and Ref. 28, Chen, A. Y., Liu, J. B., Wang, H. T., Lu, J., & Wang, Y. M. Gradient twinned 304 stainless steels for high strength and high ductility. *Mater. Sci. Eng. A* **667**, 179-188 (2016).). In this work, we studied the very local deformation behaviors of the nanotwins, and microstructures before and after tensile are given in the snapshot, and evolutions of the microstructure with the tensile strain are more fruitful than the post-mortem observations.

Reviewer's comment: *2.9. Some additional crystal plasticity simulations may be helpful to understand the transition in deformation mechanisms in NTed steel because in-situ experiments may be challenging to provide the entire picture.*

Authors' answer: That is a very good suggestion. The crystal plasticity simulation could provide the comprehensive scenario for slip activities in materials, which could be helpful for probing the deformation mechanisms in nanotwinned metals. This approach could be adopted to discuss the transition in deformation mechanisms of NTed steels in our future work. In order to reveal the deformation transition of nanotwins in ultrafine scale in this work, the dislocation-based theoretical simulations and MD simulations are utilized, and the reasonable and quantitative analysis is addressed to explain the experimental observations.

Reviewer's comment: 2.10. *The authors show that detwinning of the primary twins is dominant when external stress is closely parallel to the TBs. This is not reasonable because there is no shear stress to drive the movement of partial dislocations.*

Authors' answer: That is a very valuable question. In Fig. 2, the detwinning of primary nanotwins occurs under the case of a 9° intersection angle between the TBs of primary nanotwins and loading direction (Response figure 1a), there still has a shear component of tensile stress (Response figure 1b), which can activate the deformation modes of twinning and detwinning, as observed in Fig. 2. The dislocation-based theoretical model (in Figs. 6c and 6d) indicate that the detwinning stress is smaller than the local stress in the most observed area of $L > 28$ nm for the 60° partial system (Fig. 6c), and $L > 70$ nm for the screw partial system (Figs. 6d), thus the detwinning is the dominant deformation behavior.

Response figure 1 9° angle of loading direction to TBs of primary nanotwins.

Factually, the detwinning of the primary NTs also occurs at the condition of the TBs of primary nanotwins parallel to the loading direction. Because the TBs of the NTs with smaller λ ($\lambda < 5$ nm) can be tilted or rotated at this situation. Here, we provide two observed results: (1) In the text of Fig. 1, the TB₂ in Fig. 1a (parallel to loading direction, also shown in the Supplementary Figure S1) is titled by 6° in Fig. 1b and then by 4° in Fig. 1d after loading, then shear component is resolved.

Figure 1 | Synchronized activation of twinning and detwinning of intersected NTs.

Supplementary Figure S1. Microstructure of the intersected NTs.

(2) The MD simulation results also show that the TBs are tilted about 7° (supplementary S8a), 9° (supplementary figure S8b) and 10° (supplementary S8c) after tensile for the NT with $\lambda=5$ nm under the TB parallel to the loading direction (supplementary S8d). The same results are also confirmed in Supplementary figure S9. Therefore, a shear component still can be resolved.

Supplementary Figure S8. Snapshots of twinning and detwinning of a TB in Fe-Cr-Ni SS under uniaxial tensile deformation ($\lambda=5$ nm) showing the twinning and detwinning.

Supplementary Figure S9. Snapshots of the formation process of the TBs^{2nd} in NTed Fe-Cr-Ni SS under uniaxial tensile deformation ($\lambda=5$ nm), showing the formation of secondary twins.

Reviewer's comment: 2.11. *In-situ tests show that detwinning is followed by martensitic transformation, but is missing in the modeling.*

Authors' answer: The martensitic transformation occurs after the detwinning of nanotwins, which is similar to the martensitic transformation of the conventionally coarse austenite grains. However, in this work, we focus on the deformation behaviors of the nanotwins with different twin-spacing, and try to elucidate the physical principles through simulation. Although we used a large-scale atomic/molecular massively parallel simulator (LAMMPS) code in our MD simulations, the martensitic transformation is limited by the simulation modeling, and out of our works.

Reviewer's comment: 2.12. *What is the lattice constant chosen for the MD simulation?*

Authors' answer: As mentioned in the “Molecular dynamics (MD) simulation” section of Supplementary Note 4, the models were prepared subject to FCC structure with a lattice constant equal to 3.596 Å (as given in **Supplementary Page 20 Line 4**).

Now, we added the corresponding part in **Page 14 Line 16**: “The models were prepared to be FCC structure with a lattice constant equal to 3.596 Å.”.

Reviewer's comment: *2.13. Are the simulation results sensitive to the EAM potential used in the presented MD simulations which was designed for FeNiCr instead of SS?*

Authors' answer: Our tested specimen of 304 SS is one type of Fe-Cr-Ni stainless steels (SS), and the chemical compositions were given in Supplementary Note 2 in **Page 13 line 2**. The EAM potential selected for our MD simulations has been identified suitable for the study of microstructure evolution during the plastic deformation of Fe-Ni-Cr stainless steels (eg. Fe-20Cr-25Ni, Chen T. et al. Acta Mater. 138, 83-91 (2017). Fe-20Cr-10Ni, Terentyev D. J. Nucl. Mater. 442, 208-217 (2013)). In addition, our model of Fe-18Cr-8Ni (the typical composition of austenitic stainless steel used in the experimental tests) also repeated the experimental observations, which enhanced the credibility of our simulation results.

Reviewer's comment: *2.14. TBs could not be identified in Fig. S10. How did the authors identify twinning and detwinning partial dislocations in the simulations? Additional videos showing detwinning and twinning processes are strongly suggested.*

Authors' answer: TBs could not be identified because the atoms in Fig. S10 are colored by atom type. Also as mentioned in the “Molecular dynamics (MD) simulation” section of Supplementary Note 4, common neighbor analysis method is employed to characterize the inherent nanostructure evolution, which is widely employed to detect structural information, such as TBs. The starting structures and a video showing twinning/detwinning during uniaxial tension were added in the Supplementary figure S7, S8d, S9d, S10g and in Supplementary video figure S10.

Supplementary Figure S7. Snapshot of the NT structure before relaxation ($\lambda=10$ nm).

Supplementary Figure S8. Snapshots of twinning and detwinning of a TB in Fe-Cr-Ni SS under uniaxial tensile deformation ($\lambda=5$ nm). **a**, Partial dislocations are ready to propagate. **b**, The formation of incident TB after the propagation of b_1 and detwinning due to the propagation of a bundle of b_2 in **a**. **c**, Thickening of incident TBs owing to the propagation of b_3 in **b**. **d**, Snapshot before deformation, which is the original configuration.

Supplementary Figure S9. Snapshots of the formation process of the TBs^{2nd} in NTed Fe-Cr-Ni SS under uniaxial tensile deformation ($\lambda=5$ nm), showing the formation of secondary twins. a, Upcoming propagation of partial dislocation **b** along the extrinsic stacking fault (ESF). **b**, The formation of TBs^{2nd} along with the propagation of **b**. **c**, The final configuration of TBs^{2nd} between TBs^{1st} evolved from ESF. **d**, Snapshot of the relaxed configurations before deformation.

Supplementary Figure S10. Snapshots of twinning and detwinning process on behalf of a series of partials parallel to pre-existing TBs in NTed SS under uniaxial tensile deformation ($\lambda=2.5$ nm). a, Partial dislocation b_1 propagates parallel to pre-existing TBs. b, Detwinning in green region because of b_1 in a. c, The propagation of b_2 beside the TB leads to detwinning in pink region. d, Twinning in green region as a result of the propagation of b_3 . e, The formation of TB in green region after the propagation of b_3 in d. f, Simultaneous detwinning and twinning in green and pink regions due to the propagation of b_4 and b_5 , respectively. g, Snapshot of the relaxed configurations before deformation. At first, a partial dislocation b_1 in green region

Reviewer's comment: 2.15. Theoretical model and MD simulations are considered to be essential to the understanding of the deformation mapping presented in this work. However, most discussion has been buried in the supplementary information, which makes the related sections difficult to follow.

Authors' answer: According to the review's suggestion, we re-arranged the corresponding content in the main text. The detailed descriptions related to the principle, modeling, boundary conditions were given in the supplementary Note 3 and Note 4. While the results about the deformation features were discussed in the main text. As given in **Page 9 line 11**: "According to the theoretical model (see Supplementary Note 3), twinning is triggered under the condition of twinning stress

lower than both local stress and trailing stress. Fig. 4b gives the stress distribution of twinning, trailing, local and detwinning stresses in 60° partial system under the condition of the tensile stress parallel to the 1st TB. Only at the range of 5-149 nm, the twinning stress is lower than both the local stress (upper bound) and trailing stress (lower bound), thus, the twinning deformation can be initiated. The two critical values, transition points I and II, are very close to the *in-situ* TEM observation results. Furthermore, the theoretical results also predict that the confined range is strongly related to the orientation angle of TB with tensile stress and the types of partial dislocations. For example, the λ are 4.8-186 nm for the 60° system at 85.2° , and 6.7-237 nm for the screw system at 26.5° (see Supplementary Fig. S6), where the two orientation angles are thought to be very easy to trigger NTs in the two partial systems^{15,16}. Another point we should emphasize is that the twinning stress decreases linearly with the increase of λ , suggesting that the NTs with larger λ are easier to generate secondary twins. That is why the secondary twinning is initiated from the NT with a larger λ to the one with a smaller λ , as displayed in Fig. 3.” and in **Page 11**

Line 7: “The intrinsic mechanisms to trigger twinning/detwinning are elucidated by considering two active partials of 60° and screw systems. Although it is difficult to separate and probe the 60° dislocation and screw dislocation in TEM tests limited by the high density of pre-set dislocations³³, they are the two classic types of dislocation systems to activate the twinning/detwinning deformation^{5,16}. Furthermore, the activation of twinning/detwinning not only depends on the magnitude of twinning, trailing, detwinning and local stress, but also is influenced by the orientation angle of TBs with loading direction. Fig. 6 gives the stress distribution of the two partial systems with two orientation angles of 70° (a, b) and 9° (c, d), respectively, as observed in *in-situ* TEM observations. The NT with $\lambda=2.5$ nm is selected as a typical scale, and the discussed zone of unit length (L) is 100-500 nm according to the visual sight of the TEM observations. When the twinning stress is smaller than both the trailing stress and local stress, twinning is initiated. Similarly, the detwinning stress is smaller than the local stress, then detwinning is active. In Fig. 6a, the twinning stress is smaller than the trailing stress at $L < 160$ nm, while larger than the trailing stress in

the most observed area ($L > 160$ nm), suggesting that twinning of 60° partial system occurs with a small probability.”.

Reviewer's comment: *2.16. The paper has other presentation issues. All figure captions in the main text are extremely oversimplified and should contain sufficient information so that the reader can understand the figure without frequently referring to the main text.*

Authors' answer: We had revised all the figure captions in order to give sufficient information for the readers to understand the descriptions. For example, we listed the figure caption of Figure 1 as follows:

Figure 1 | Synchronized activation of twinning and detwinning of intersected

NTs. **a-d** are the bright-field TEM images of the intersected NTs, showing the deformation evolution during *in-situ* tensile. The TB₁ exhibits a 70° angle to the tensile stress (F), and the TB₂ is parallel to F, as given in **a**. The red solid lines and arrows, such as Lines 1-3 and Arrows 1-3 in tensile direction, Line 4 and Arrow 4 in the 70° shear direction, indicate the twinning positions. While the blue dotted lines and arrows, such as Lines 2, 3 and Arrows 2, 3 in tensile direction, and Lines 5-7 and Arrows 5-7 in the 70° shear direction, illustrate the detwinning positions. Twinning and detwinning are co-activated in the NTs with λ smaller than 5 nm (with the error of 0.5 nm). All the TEM images have the same scale bar. The time frame from **a-d** is 0, 12, 27 and 52 s, respectively.

Reviewer's comment: 2.17. The writing of the manuscript is careless (e.g., page 8 line 150).

Authors' answer: We had corrected the errors in the manuscript carefully, including the sentence in **Page 8 Line 150**. It is changed from “Fig. 4b gives the stress distribution of twinning, trailing and detwinning stresses in 60° partial system under the condition of the external force parallel to the 1st TB and .” into “**Fig. 4b gives the**

stress distribution of twinning, trailing, local and detwinning stresses in 60° partial system under the condition of the tensile stress parallel to the 1st TB.” in the revised manuscript of **Page 9 Line 13**.

Reviewer's comment: *2.18. To this referee, this work does not provide necessary microscopy information, statistical analysis, and computational data to support the conclusion. Therefore, it is not suitable for publication in Nature Communications.*

Authors' answer: In the revised manuscript, we had added the necessary statistical analysis (Response 1.1, 2.8), and computational data (Response 2.14) to support the conclusions according to the reviewer's suggestions. The microscopy information is given in the *in-situ* TEM observations, while the other features, including the twin size distribution, phase transformation, dislocation density, and post-mortem observations after tensile fracture, had been reported in our previous works (Refs. 33, 28). Now, we thought we had provided enough microstructure information and computational data to reveal deformation transition mechanisms.

Reviewer #3 (Remarks to the Author):

Reviewer's comment: *In this work, the authors presented a deformation mechanism map of nanotwinned austenitic stainless steel using in-situ TEM tensile tests, dislocation-based theoretical modeling, and MD simulations. With twin spacing as the controlling parameter, the deformation map shows that the FCC stainless steel deforms through co-activated twinning/detwinning at twin spacing less than 5 nm, secondary twinning at twin spacing 5-129 nm, and dislocation glide at twin spacing larger than 129 nm. While this map is important to understand the deformation mechanism of nanotwinned materials, the authors failed to prove that this map is indeed what they claimed to be due to the many issues discussed in detail below.*

3.1. My major concern is related to the twin spacing measurement. In Figures 1, 2, and S1, it seems to me that the authors only measured the twin spacing of the very thin twin laminates to make the sample lies in the region of less than 5 nm. In fact,

the large white areas in Figures 1, 2, S1 (a) indicate that much larger twins exist with a twin spacing of about 20 nm based on my rough measurement. In addition, the authors also only measured the white line in the dark-field TEM shown Figure S1 (c), without considering the large dark areas. If my understanding is correct, this sample cannot be considered as having only twin spacing less than 5 nm. In contrast, the authors measured all the areas in the sample containing twin spacing 5-129 nm, as shown in Figure 3 (a). It sounds to me that the authors were trying to divide the two samples into different categories by using different measuring methods.

Authors' answer: Factually, we used the same method (as given in Fig. 3a) to measure the twin spacing of all the areas. As questioned by the reviewer, the large white areas in Figs. 1, 2 and S1a indicate that many larger twins exist with a twin spacing of about 20 nm. That is true. The reason we focus on the very thin twin laminates is that only the very thin twins occur the co-activation of twinning and detwinning behaviors. The statistic results (as given in Response 1.1) show that only those nanotwins with twin-spacing smaller than 5 nm are active for the co-activation of twinning and detwinning.

For these larger nanotwins (approximate $\lambda=5-40$ nm) in Fig. 1, secondary twinning are hard to be resolved. There are two TB orientations, TB₁ and TB₂, as shown below (Response figure 3a). When the secondary nanotwins are initiated in 1st nanotwins, the orientations of the newly formed 2nd nanotwins are illustrated in Response figure 3b. The orientation of 2nd nanotwins in 1st nanotwins (TB₁ direction) is along the TB₂ direction, and vice versa. Thus, there still have two twin orientations. When the 1st nanotwins occur twinning and detwinning in the TB₁ and TB₂ directions, it is hard to resolve which one is the 1st and 2nd nanotwins.

Response figure 3 70° angle of loading direction to TBs of intersected nanotwins. The newly formed 2nd nanotwins also exhibit the two TB orientations, TB₁ and TB₂.

Reviewer's comment: 3.2. *When discussing the deformation mechanism for twin spacing between 5 and 129 nm, the authors only used one sample containing a lot of twins, with twin spacing varying from 5 to 129 nm. However, the authors meant to conclude that the deformation mechanism (secondary twinning) happened in this sample will happen in the nanotwinned samples as long as their twin spacing is between 5 and 129 nm. In other words, a sample containing only 6 or 7 (or any number between 5 and 129) nm twins will deformation through secondary twinning. This can be evidenced by their dislocation-based theoretical modeling and the MD simulations, where only one twin spacing is included each time. The authors need to provide more experimental evidence to support their conclusion by doing in-situ TEM tensile tests on more samples containing different twin spacing.*

Authors' answer: The transition points of lower and upper bounds are based on the *in-situ* observations of four sample with 14 observed area, the statistic results are added in Response 1.1 (here, we provide two supplementary figures, S2 and S4). We observed secondary twinning in four samples, and the statistic results indicate that the twin spacing ranges from 6 to 129 nm, as a given example in Fig. 3. The upper bound of twin spacing might be larger than 129 nm, which is limited by the observation view of *in-situ* TEM and the evaluated error (the NTs in this work are deformation NTs, which have many kinks and steps at TBs, resulting in a varied twin-spacing.). The dislocation-based theoretical model calculates a successive distribution of twin spacing (Fig. 4b), and derives a transition range of 5-149 nm for 60° partial system.

The MD simulations (twin spacing of 2.5, 3.74, 5, 10, 15 nm) are carried out using only one twin spacing each time, limited by the calculation capacity. We provide three other *in-situ* examples to show that secondary twinning occurs in nanotwins with multiscale scale, as shown in the following examples.

Example 1:

Response figure 4 Dark-field TEM of the *in-situ* tensile tests. The red arrows indicate the twinning deformation, and the blue arrow show the detwinning deformation of nanotwins with twin spacing of 14 and 25 nm.

Example 2:

Response figure 5 Bright-field TEM of the *in-situ* tensile tests. The red arrows indicates the twinning deformation, and the blue arrow show the detwinning deformation of the nanotwins with twin spacing of 37 and 55 nm.

Example 3:

Response figure 6 Bright-field TEM of the *in-situ* tensile tests. The red arrows indicate the twinning deformation, and the blue arrow show the detwinning deformation of the nanotwins with twin spacing of 38 and 35 nm.

The added parts are given in **Page 5 Line 8**: “The statistic results show that co-activation of twinning and detwinning occurs in the intersected and primary NT bundles with λ smaller than 5 nm (with evaluated error of 0.5 nm) under shear stress, and the NTs with $\lambda=2-3$ nm contribute the highest ratio of 45% (see Supplementary Fig. S2).” and **Page 7 Line 17**: “The statistic results exhibit that 80% NTs with $\lambda < 5$ nm (5 observed zones) exhibit the co-activation of twinning and detwinning under shear stress, including intersected and primary NTs. While 100% NTs with $\lambda < 5$ nm (2 observed zones) exhibit detwinning and the subsequent martensitic transformation when the TBs are closely parallel to the tensile direction. Secondary twinning occurs in primary NTs ($\lambda=6-129$ nm) with a frequency of 80% among the 5 observed zones. Only dislocation motion is observed in the left 2 observed zones, where the λ of those twins are at submicrometer scale. The statistic diagram is given in Supplementary Fig. S4.” in **Page 14 Line 3**: “Four samples with 14 micro-zones were observed through *in-situ* TEM experiments, where 7 observed zones had the NTs with $\lambda < 5$ nm, 5 observed zones were the NTs with $5 \text{ nm} < \lambda < 129$ nm, and 2 observed zones were the twins with λ at submicrometer scale.”.

Supplementary Figure S2. Statistic distribution of λ characterized by co-activation of twinning and detwinning in the intersected NTs. The NTs with $\lambda < 5$ nm are *in-situ* observed co-activation of twinning and detwinning, and the NTs with $\lambda=2-3$ nm contribute the highest ratio. The statistical errors of the evaluated λ are 0.5 nm.

Supplementary Figure S4. The statistic diagram of the deformation behaviors.

14 micro-zones are observed through *in-situ* TEM experiments, where 7 observed zones have the NTs with $\lambda < 5$ nm, 5 observed zones are the NTs with $5 \text{ nm} < \lambda < 129$ nm, and 2 observed zones are the twins with λ in ultrafine scale, as given in **a**. The determined errors of λ are 20 nm for upper limit of 129 nm and 1 nm for the lower limit. The statistic results exhibit that 80% NTs (5 observed zones with $\lambda < 5$ nm) are co-activated twinning and detwinning under 70° angle of TB to loading direction. 100% NTs (2 observed zones with $\lambda < 5$ nm) exhibit detwinning and the subsequent martensite transformation under 9° angle of TB to loading direction. Among the 5 observed zones containing NTs with $\lambda = 5$ -129 nm, secondary twinning occurs with a frequency of 80%. In the last 2 observed zones, dislocation motion is active in the NTs with λ in ultrafine scale.

Reviewer's comment: 3.3. Only 60-degree and screw dislocations were considered in the dislocation-based theoretical model to explain the effect of twin spacing on the twinning, detwinning, etc stress and thus on the change of deformation pattern. But the authors did not mention anything that these two kinds of dislocations are

involved in the TEM tests. The theoretical modeling told nothing if such kinds of dislocations did not happen. It is of course highly possible that these dislocations are involved, but the authors at least should say something to make the connection between theoretical modeling and TEM tests. In addition, it should not be just the 60-degree dislocation and screw dislocations that are responsible for the twinning, detwinning etc. Other dislocation types such as edge dislocation, mixed dislocation should also be considered in the theoretical modeling. If by any chance there are technical difficulties dealing with these dislocations, the authors should at least discuss the situation in their modeling work.

Authors' answer: The reviewer's comment is right that the theoretical model only considers the 60-degree and screw dislocations, which are not mentioned in TEM tests. As pointed out in Ref. [Zhu et al., J. Appl. Phys. 98, 034319, 2005], the 60-degree dislocation system and screw dislocation system are two kinds of classic dislocation system for deformation twinning, while the dislocation in a grain is most likely a mixed nature. Therefore, these two kinds of classic dislocation systems are difficult to be separated and probed in TEM tests. The corresponding discussions have been added in the manuscript **Page 11 Line 9**: “Although it is difficult to separate and probe the 60° dislocation and screw dislocation in TEM tests limited by the high density of pre-set dislocations³³, they are the two classic types of dislocation systems to activate the twinning/detwinning deformation^{5,16}. Furthermore, the activation of twinning/detwinning not only depends on the magnitude of twinning, trailing, detwinning and local stress, but also is influenced by the orientation angle of TBs

with loading direction.” and in Supplementary Note 3 **Page 17 Line 2**: “Here, it should be pointed out that the 60° dislocation system and screw dislocation system are two kinds of classic dislocation system for deformation twinning. While the dislocation in a grain is most likely a mixed nature [S2], therefore, it is difficult to separate and probe the 60° dislocation and screw dislocation in TEM tests during deformation twinning.”.

The reviewer’s comment is also correct that the 60-degree dislocation system and screw dislocation system are involved in the model and the other types of dislocations are not considered in the present model. Actually, based on the Thompson tetrahedron illustrating the possible slip planes in FCC crystal as discussed in Ref. [Zhu et al., Acta Mater. 59, 812-821, 2011], only the mixed dislocation and screw dislocation contribute to the deformation twinning. The 60-degree dislocation system refers to the mixed dislocations. The corresponding statements have been added in Supplementary Note 3 **Page 14 Line 15**: “According to the Thompson tetrahedron illustrating the possible slip planes in fcc crystal, the mixed dislocation and screw dislocation contribute to the deformation twinning [S3]. The 60° dislocation system is associated with the mixed dislocations.”.

Reviewer's comment: 3.4. In the dislocation-based modeling, the authors defined the 60-degree dislocation as the case when $\alpha_1=90$ and $\alpha_2=30$. That way the angles between the Burgers vectors of the leading and trailing partial would be $90+30=120$, which seems against my understanding. For fcc metals, the angle between the two Burgers vectors should always be 60 degrees, independent of the dislocation type.

Authors’ answer: The reviewer’s comment is quite true that the angle between the two Burgers vectors should be 60 degrees for all types of dislocations. In the proposed model, we only define the angles, α_1 and α_2 , but lost the sign of the angles. It should be noted that the sign of the angle with the Burgers vector moves

anticlockwise from partial dislocation line is positive, and the opposite is negative. Therefore, the signs of the angles α_1 and α_2 in the 60-degree dislocation system are both positive, leading to the intersection angle between α_1 and α_2 is 60 degrees. For the screw dislocation systems, $\alpha_1 = -\alpha_2 = 30$ degree. The corresponding modification has been made in Supplementary Note 3 **Page 14 Line 12**: “As shown in Fig. S12, when $\alpha_1 = 90^\circ$ for leading partial and $\alpha_2 = 30^\circ$ for trailing partial, such dislocations are called as the 60° system. When $\alpha_1 = 30^\circ$ for leading and $\alpha_2 = -30^\circ$ for trailing partials, these dislocations are defined as the screw system [S2].”, and in Supplementary Note 3 **Page 15 Line 19**: “Let’s define the angle to be positive when the angle rotates anticlockwise from the dislocation line.”.

Reviewer's comment: 3.5. *The TEM images in Figure S3 is not clear. Although the authors claim this is an image for a sample containing twins in submicrometer scale, it seems to me that this image contains a twin with a spacing of about 100 nm, less than 129 nm.*

Authors’ answer: That is right. There is a twin (Twin I) with a spacing of 118 nm in the Supplementary Figure S5, while there are two other twins (Twin II and III) with twin spacing at the submicrometer scale, as shown Supplementary Figure S5 below. The dislocation motions are observed in the three twins. Additionally, secondary twinning is also observed in the Twin I (a primary twin with a twin spacing of 118 nm), as indicated in figure S5b. Because the TB orientation of these secondary nanotwins is closely parallel to the loading direction, the detwinning of 2nd nanotwins is dominant at this loading condition, as indicated in Figure S5c.

Supplementary Figure S5. The formation of dislocation walls (DWs) at the TBs with λ at the submicrometer scale.

Reviewer's comment: 3.6. *In Page 11, lines 208-212, the authors claimed that detwinning is the dominant deformation mechanism for the screw dislocation. This is however inconsistent with Figure 6d showing that detwinning stress is highest in the whole zone and thus detwinning is least likely.*

Authors' answer: The activation of detwinning depends on two factors. One is that there have pre-set nanotwins, the other is that detwinning stress should be smaller than the local stress independent of the trailing stress or twinning stress. In Fig. 6d, since the detwinning stress is smaller than the local stress at $L > 60$ nm, the detwinning of the existing nanotwins is triggered, which occupies the majority of the observed area ($100 \text{ nm} < L < 500 \text{ nm}$). Similarly, the twinning is activated in $18 \text{ nm} < L < 180 \text{ nm}$ since the twinning stress is smaller than the trailing stress and local stress, while twinning is prohibited in $180 \text{ nm} < L < 340 \text{ nm}$ (most of the observed area) due to the trailing stress larger than the twinning stress. Thus, for the screw system at the 9° intersected angle of loading direction with TB, detwinning is dominant, and twinning is also activated but with a relatively small area ($100 \text{ nm} < L < 180 \text{ nm}$). Final, the nanotwins disappear due to the overwhelming majority of detwinning compared to twinning.

In order to clarify the ambiguous statements, we revise the corresponding content in Page 11, lines 208-212, as given in **Page 11 Line 19**: “When the twinning stress is smaller than both the trailing stress and local stress, twinning is initiated. Similarly, the detwinning stress is smaller than the local stress, then detwinning is active.” and in **Page 12 Line 14** “In the case of 9° angle, the twinning is very hard to initiate for the 60° system due to the twinning stress larger than the trailing stress (Fig. 6c), while the detwinning is the most possible deformation mode resulting from the smaller detwinning stress. However, for the screw system in Fig. 6d, the twinning is only activated at $18 \text{ nm} < L < 180 \text{ nm}$. This means that the twinning has a small probability since twinning is prohibited in the most of the observed area ($180 \text{ nm} < L < 340 \text{ nm}$). However, the detwinning of the existing NTs is triggered in the whole observed area ($L > 60 \text{ nm}$). Thus, for the screw system at the 9° intersected angle of TB to loading direction, detwinning is the dominant deformation behavior in the observed area, and

twinning is also activated but with a small frequency. Final, the NTs disappear due to the overwhelming majority of detwinning compared to twinning.”.

Reviewer's comment: 3.7. *What is the crystal orientation perpendicular to the paper for the TEM images? Without this orientation, the angle between the applied force and the TB orientation is not well-defined and it is not appropriate to use the theoretical model and MD simulations to explain experimental results.*

Authors' answer: The zone axes of the intersected NTs in Fig. 1a are [1-10] (70° angle with loading direction) and [-110] (parallel to loading direction), respectively. The similar intersected angles between the loading direction and the TB orientation are used in the theoretical model and MD simulations (see the following figure 5 in the text and figure S8 in the supplementary information) to explain experimental results.

Figure 5 | Snapshots of twinning and detwinning of TBs^{2nd} in 1st NT under uniaxial tensile deformation ($\lambda=10$ nm).

Supplementary Figure S8. Snapshots of twinning and detwinning of a TB in Fe-Cr-Ni SS under uniaxial tensile deformation ($\lambda=5$ nm).

Reviewer's comment: 3.8. *Exactly what did the authors mean by “external stress,” “horizontal stress,” “external stress” etc? Are they tensile stress or shear stress.*

Authors' answer: The external stress is the tensile stress, horizontal stress also is the tensile stress, which is applied in the horizontal direction. Shear stress is a shear component of the tensile stress when tensile stress exhibits a 70° angle to the TBs. In order to clarify all these stress types, we replace “external stress” into “tensile stress”, and “horizontal stress” into axial stress, and still remain the “shear stress” in the manuscript and supplementary information.

Reviewer's comment: 3.9. *What did the authors mean by “incident twin boundary (ITB)” in Figure S1? And which TB is CTB, which TB is ITB?*

Authors' answer: The incident twin boundary (ITB) is the top or the end of the TB, while the coherent twin boundary (CTB) is the body part of the TBs, as illustrated below.

Generally, the twinning or detwinning includes the growth or shrinkage of the TBs from the ITB, and new formation of nanotwins with CTB. In order to avoid confusion, we delete the corresponding ITB and CTB parts in the Supplementary Note 1 (**Page 1 Line 12**): “The high-resolution TEM (HRTEM) image (Supplementary Fig. S1d) displays the NTed microstructure of the rectangular zone in Fig. S1a.”.

Reviewer's comment: 3.10. *What does the red shaded area mean in Figure 6?*

Authors' answer: The red shaded area represents the observed area of the *in-situ* TEM view ($L=100-500$ nm) during tensile. According to the reviewer's suggestion (3.12), we delete this illustrations, and revise the Fig. 6 into the following format:

Figure 6 | Distribution of twinning, detwinning and trailing stresses during tensile deformation. a,b, 70° angle of the 1st TB to the tensile direction. c,d, 90° angle of the 1st TB to the tensile direction. a,c, 60° system. b,d, screw system. The activation of twinning requires the twinning stress lower than both the local stress and trailing stress, thus, twinning is prohibited at the unit length (L) larger than 180 nm in a. When the detwinning stress is smaller than the local stress, detwinning is initiated. Therefore, detwinning is dominant in the most of area ($L > 40$ nm) in a. On the opposite, twinning is the featured deformation behavior due to the negative trailing and detwinning stresses in screw system in b. Detwinning controls the deformation in c ($L > 40$ nm) and d ($L > 60$ nm), while twinning also occurs in d with a smaller range ($L < 180$ nm).

Reviewer's comment: 3.11. Page 10, lines 186, the authors claimed that the twinning/detwinning in this work is very special. What did they mean by “very special?”

Authors' answer: Detwinning has been reported in many works [Refs. 7 and 8], and it becomes one of the softening mechanisms of materials. However, we found

successive twinning and detwinning in ultrafine nanotwins (twin-spacing smaller than 5 nm), which could sustain the work hardening by alternative twinning and detwinning. Co-activation of twinning and detwinning deformation is special, which might occur in other FCC NTed metals. The added part is given in **Page 11 Line 5**: “The detwinning of NTs with much finer λ is commonly observed^{7,8}, however, the co-activation of twinning and detwinning in this work is very special, which should be very helpful for the improvement of plasticity and strength.”.

Reviewer's comment: 3.12. Page 10, lines 193–201. When the authors discussed the zone effect on twinning/detwinning, why the authors gave an upper or lower bound in the discussed zone. For example, that the trailing stress is smaller than the twinning stress at 160-500 nm length, is also true for the length larger than 500 nm. Then why the authors limited the zone to 160–500 nm, instead of just >160 nm. Similarly, 100-160 nm can also be <160 nm, 100-370 nm can be <370 nm, etc.

Authors' answer: The pre-set upper and lower bounds (L=100-500 nm) are according to the *in-situ* TEM observation view (as we given in **Page 11 line 18**), which are initially used to reflect which one is dominant or which one has the high probability between twinning and detwinning deformations. For example, in Fig. 6a, the trailing stress is larger than the twinning stress at 160-500 nm, and smaller than the twinning stress at 100-160 nm, therefore, the twinning only is initiated in a relative short range of 100-160 nm, while detwinning is activated at a large length of 160-500 nm. Thus, we think detwinning is the dominant behavior in 60° partial system.

Now, we accept the reviewer' suggestions. We cancel the confined area of upper and lower bounds, which indeed do not influence the conclusions. For example, the revised part in **Page 11 Line 15** is that: “Fig. 6 gives the stress distribution of the two partial systems with two orientation angles of 70° (a, b) and 9° (c, d), respectively, as observed in *in-situ* TEM observations. The NT with $\lambda=2.5$ nm is selected as a typical scale, and the discussed zone of unit length (L) is 100-500 nm according to the visual sight of the TEM observations. When the twinning stress is smaller than both the

trailing stress and local stress, twinning is initiated. Similarly, the detwinning stress is smaller than the local stress, then detwinning is active. In Fig. 6a, the twinning stress is smaller than the trailing stress at $L < 160$ nm, while larger than the trailing stress in the most observed area ($L > 160$ nm), suggesting that twinning of 60° partial system occurs with a small probability. However, the detwinning stress is lower than the local stress in the observed area ($L > 40$ nm), thus, the detwinning is dominant. Opposite to the 60° partial system, the trailing stress of screw system is negative at $L < 370$ nm, and larger than the twinning stress at $L > 370$ nm, therefore, twinning occurs in the whole observed area (Fig. 6b).”.

Reviewer's comment: 3.13. *In MD simulations, what do the sample configuration and twin structure looks like before deformation? The authors only gave the snapshot after deformation, which is clearly not enough to help understand the simulation results.*

Authors' answer: The *in-situ* specimen produced by surface mechanical attrition treatment with a high impacting speed is composed of deformed nanotwins, containing multiscale nanotwins and defected structure. As the reviewer queried, in MD simulations, we relaxed the sample configuration similar to the real structure in the *in-situ* tensile, in order to reveal the deformation evolution of the primary and secondary twins under loading. According to the comments of the reviewer, we added all the relaxed configurations before deformation in the Supplementary information, including Supplementary Figure S8-10. The revised Figures were given below:

Supplementary Figure S8. Snapshots of twinning and detwinning of a TB in Fe-Cr-Ni SS under uniaxial tensile deformation ($\lambda=5$ nm). **a**, Partial dislocations are ready to propagate. **b**, The formation of incident TB after the propagation of b_1 and detwinning due to the propagation of a bundle of b_2 in **a**. **c**, Thickening of incident TBs owing to the propagation of b_3 in **b**. **d**, Snapshot before deformation, which is the original configuration.

Supplementary Figure S9. Snapshots of the formation process of the TBs^{2nd} in NTeD Fe-Cr-Ni SS under uniaxial tensile deformation ($\lambda=5$ nm), showing the formation of secondary twins. **a**, Upcoming propagation of partial dislocation **b** along the extrinsic stacking fault (ESF). **b**, The formation of TBs^{2nd} along with the propagation of **b**. **c**, The final configuration of TBs^{2nd} between TBs^{1st} evolved from ESF. **d**, Snapshot of the relaxed configurations before deformation.

Supplementary Figure S10. Snapshots of twinning and detwinning process on behalf of a series of partials parallel to pre-existing TBs in NTed SS under uniaxial tensile deformation ($\lambda=2.5$ nm). a, Partial dislocation b_1 propagates parallel to pre-existing TBs. b, Detwinning in green region because of b_1 in a. c, The propagation of b_2 beside the TB leads to detwinning in pink region. d, Twinning in green region as a result of the propagation of b_3 . e, The formation of TB in green region after the propagation of b_3 in d. f, Simultaneous detwinning and twinning in green and pink regions due to the propagation of b_4 and b_5 , respectively. g, Snapshot of the relaxed configurations before deformation. At first, a partial dislocation b_1 in green region

Reviewer's comment: 3.14. Page 11, lines 222-223. *The authors claimed that their conclusion should be generally applicable to other NTed fcc metals. This is actually inconsistent with their claim on Page 10, line 186, that their “successive twinning/detwinning found in this work is very special.”*

Authors' answer: As we discussed above (Response 3.11), detwinning has been reported in many works, and it becomes one of the softening mechanisms of materials. However, we found successive twinning and detwinning in ultrafine nanotwins (twin-spacing smaller than 5 nm), which can sustain the work hardening by alternative twinning and detwinning. That is different from the only one deformation mode of detwinning.

By using *in-situ* TEM tensile, successive twinning and detwinning were observed in FCC austenitic stainless steel, it might be also active in other FCC NTed metallic materials when the twin-spacing is smaller than a critical value. Co-activation of twinning and detwinning deformation are special, which might occur in other FCC NTed metals.

Reviewer's comment: 3.15. *The title is confusing. I don't understand why the authors used the word "holographically" in their title. It seems to me that the authors were trying to say that their map contains more information than the usual map probably mentioned in the literature work. Even this is the case, the word is still confusing and ambiguous.*

Authors' answer: We accepted the reviewer's suggestion and revised the title into "Mapping the deformation behaviors of nanotwins: scale law of complex deformation transitions".

Reviewer's comment: 3.16. *Page 5, line 88, what does "one-system" mean here?*

Authors' answer: The phrase, "one-system", means "one-order or primary", where the "one-system NT bundles" means "primary NT bundles" in the text. For clearness, we have revised the corresponding descriptions into "primary NT bundles" in **Page 5 Line 10**.

Reviewer's comment: 3.17. *Page 6, line 112, why there is 5 nm twin in nanotwins with spacing in the range of 6-129 nm?*

Authors' answer: Here, we *in-situ* observed the deformation behaviors of five nanotwins (NT 1, 2, 3, 4, 5) with twin-spacing of 129, 42, 15, 6 and 5 nm. The *in-situ* TEM results exhibited that the secondary twinning was generated inside the NTs (NT 1-4) with $\lambda = 6-129$ nm, and not initiated in the NT 5 with $\lambda=5$ nm. Therefore, we suggested that the secondary twinning occurred in NTs with λ in the range of 6-129 nm. And then the $\lambda = 5$ nm was used as a transition point from co-activated twinning/detwinning ($\lambda < 5$ nm) to secondary twinning ($5 \text{ nm} < \lambda < 129$ nm).

Reviewer's comment: 3.18. *Page 7, line 136, is "dislocation motivation" actually "dislocation activation?" The authors used "dislocation motivation" in a lot of places in the main text and supplementary material.*

Authors' answer: As the suggestion of the reviewer, the word "dislocation motivation" had been changed into "dislocation motion" in **Page 8 line 1, Page 8 line**

10, and in Supplementary Page 5 line 11.

Reviewer's comment: 3.19. Page 12, line 236. A reference is needed for the LAMMPS code and the EAM potential for Fe-Ni-Cr.

Authors' answer: As the suggestion of the reviewer, we have added the references in the revised manuscript, as shown in Page 14 Line 10: “MD simulations (see Supplementary Note 4) were carried out using the large-scale atomic/molecular massively parallel simulator (LAMMPS) code ⁴¹. An embedded atom method potential for Fe-Ni-Cr system was employed to describe the atomic interaction in the sample ^{42,43}”.

The added references are listed below:

41. Plimpton, S. Fast parallel algorithms for short-range molecular dynamics. *J. Comput. Phys.* **117**, 1-19 (1995).
42. Chen, T., Tan, L., Lu, Z., & Xu, H. The effect of grain orientation on nanoindentation behavior of model austenitic alloy Fe-20Cr-25Ni. *Acta Mater.* **138**, 83-91 (2017).
43. Bonny, G., Terentyev, D., Pasianot, R.C., Poncé, S., & Bakaev, A. Interatomic potential to study plasticity in stainless steels: the FeNiCr model alloy. *Modelling Simul. Mater. Sci. Eng.* **19**, 085008 (2011).

Reviewer's comment: 3.20. There are a lot of grammar issues in this manuscript. For example, Page 3, line 40, “dependence” should be “dependent;” Page 8, line 144, “predict” should be “predicted;” Page 10, line 188, “elucidate” should be “elucidated” etc.

Authors' answer: The descriptions have been carefully corrected. The words “dependence” has been changed into “dependent” in Page 3 line 1, and “predict” into “predicted” in Page 9 line 9, “elucidate” into “elucidated” in Page 11 line 8.

Reviewer's comment: 3.21. Overall, this manuscript is not well-prepared and, as a result, the story is not well-presented. I can get what the authors were trying to

prove but I am confused about so many details. On one hand, the authors should provide more evidence to prove their theory; on the other hand, they should also try to draft this paper in a clear and lucid style.

Authors' answer: We had carefully re-organized the manuscript, and provided more evidences to prove our results. Now, we confirm this text is clear and lucid style.

Many thanks to both the editor and the referees for their very helpful suggestions on this manuscript!

Best regards!

A.Y. Chen

Reviewers' comments:

Reviewer #1 (Remarks to the Author):

The authors revealed the relationship between the twin size and the deformation mechanism through in-situ TEM experiments, theoretical analysis and MD simulations and built up a comprehensive deformation pattern in the stainless steel. These results in the manuscript may serve as a significant guidance for understanding the deformation behaviors in nano-twinned metals and designing advanced structural materials with excellent mechanical performances.

However, "dependence of the deformation mechanism on the twin size in nano-twinned metals" is not a new concept. This topic has been investigated a lot in low stacking faults energy metals.

Moreover, some revisions should be done listed below:

1. The in-situ TEM image showed in this manuscript were highly consistent with the simulation results. Could the authors provide ballpark fractions about nano-twins with different sizes corresponding to different deformation mechanisms in the whole observed area? I guess the statistic results might be more solid to support the conclusion rather than the single example.
2. The nucleation of martensite after the detwinning mentioned in the results. Could the authors give some discussion about the internal connection between them? It might also be a good topic to explore in the future work.
3. The detwinning might lead to the softening in metals. Could the authors add some discussion about the strength-softening effects from different deformation mechanisms?
4. There are some errors in the reference list like the authors in the reference [2], please re-check that.

Reviewer #2 (Remarks to the Author):

The authors have addressed some concerns from the previous review. But the referee is still concerned about the novelty of the present paper. The authors stated that essential microstructure characterization of NTed SS has already been reported in their previous work (Chen, A. Y., Ruan, H. H., Wang, J., Chan, H. L., Wang, Q., Li, Q., & Lu, J. The influence of strain rate on the microstructure transition of 304 stainless steel. *Acta Mater.* 59, 3697-3709 (2011)). They also stated that the mechanical properties of NTed SS have been explored in their previous papers (Chen, A. Y., Liu, J. B., Wang, H. T., Lu, J., & Wang, Y. M. Gradient twinned 304 stainless steels for high strength and high ductility. *Mater. Sci. Eng. A* 667, 179-188 (2016), A.Y. Chen, W.F. Hu, et al., Improving the intergranular corrosion resistance of austenitic stainless steel by high density twinned structure, *Scripta Materialia* 130 (2017) 264-268). It seems that various deformation mechanisms of NTed SS have also been proposed and discussed in their pervious papers to rationalize the improved mechanical properties of NTed SS. Overall, to this referee, the present paper looks more like an incremental progress instead of an important advance of significance on the understanding of deformation behaviors of NTed SS.

In view of its lack of sufficient novelty, this paper does not meet the standard of Nature Communications. A more specialized journal (e.g., *Acta Materialia*) might be a better choice.

Reviewer #3 (Remarks to the Author):

The authors have addressed most of my comments. From their response, I realized why I was so confused about the way they defined the deformation map based on twin boundary spacing. Set by the literature work on nanotwin materials, I have the tendency to think each tested sample should contain only one region. Instead, it seems that the sample in this work can contain twins with much different sizes. The deformation mechanism of different regions in this sample depends on the twin spacing in this region. In other words, a nanotwinned sample can deform simultaneously through all of the mechanism proposed in this work. If my understanding is right, the authors should make this point clear in the abstract to avoid this kind of confusion to the prospective readers.

Response to the Reviewers' Comments

Ref. No.: NCOMMS-18-14200B

Title: Scale law of complex deformation transitions of nanotwins in stainless steel

Authors: A. Y. Chen, L. L. Zhu, L. G. Sun, J. B. Liu, H. T. Wang, X. Y. Wang, J. H. Yang, J. Lu

Reviewers' comments:

Reviewer #1 (Remarks to the Author):

Reviewer's comment: *The authors revealed the relationship between the twin size and the deformation mechanism through in-situ TEM experiments, theoretical analysis and MD simulations and built up a fairly comprehensive deformation pattern in the stainless steel. These results in the manuscript may serve as a significant guidance for understanding the deformation behaviors in nano-twinned metals and designing advanced structural materials with excellent mechanical performances.*

However, "dependence of the deformation mechanism on the twin size in nano-twinned metals" is not a new concept. This topic has been investigated a lot in

low stacking faults energy metals.

Authors' answer: As pointed out by the reviewer, it is right that the dependence of twin size on the deformation mechanism in nano-twinned metals has been investigated a lot in low stacking fault energy metals. These studies, especially based on the growth nanotwins in Cu, Ag, and Ni metals, focus on the effect of nanotwins on the mechanical properties, and the corresponding strengthening mechanisms. The research works exhibit complicated deformation behaviors of nanotwins, involving the dislocation pile-up at TBs, partial dislocation sliding along TBs, and TB migration, or detwinning. However, the underlying deformation mechanisms of twin boundaries (TBs) in nanotwins are still unclear, limited by the multiscale (twin-lamella-spacing, λ , from several nanometers to hundreds of nanometers) and multiform (primary, secondary, and hierarchical systems) microstructure features. The leading factor to trigger these deformation mechanisms is ambiguous. Therefore, the construction of deformation map becomes a key issue.

In this work, the nanotwins in structural steels are the deformation nanotwins, which possess high density of defective structures. These defective structures not only control the deformation of one-order nanotwins, but also form the two-order nanotwins. We in-situ observe the deformation behaviors of those defective twin boundaries and build up a fairly comprehensive deformation pattern to clarify the twin-lamella-dependent deformation mechanism in nanotwinned metals. Based on the direct evidences, two deformation transitions are found experimentally from co-activated twinning/detwinning ($\lambda < 5$ nm) to secondary twinning (5 nm $< \lambda < 129$

nm), and then to dislocation glide ($\lambda > 129$ nm). The intrinsic transition mechanisms were revealed by a dislocation-based theoretical model and molecular dynamics simulations.

Moreover, some revisions should be done listed below:

Authors' answer: The following suggestions of the reviewer have been revised and added in the first-round response. Now, we list the corresponding parts below:

1.1. The in-situ TEM image showed in this manuscript were highly consistent with the simulation results. Could the authors provide ballpark fractions about nano-twins with different sizes corresponding to different deformation mechanisms in the whole observed area? I guess the statistic results might be more solid to support the conclusion rather than the single example.

Authors' answer: That is a very valuable suggestion to give a statistic analysis of deformation features. Factually, we carried out four samples in *in-situ* TEM tests, and 14 micro-zones were observed. The results we gave in the manuscript were the dominant observations. The corresponding descriptions of the samples were added in **Page 13 Line 17**: “Four samples with 14 micro-zones were observed through *in-situ* TEM experiments, where 7 observed zones had the NTs with $\lambda < 5$ nm, 5 observed zones were the NTs with $5 \text{ nm} < \lambda < 129$ nm, and 2 observed zones contain the twins with λ at submicrometer scale.”.

The detailed statistic results of the deformation features are given in **Page 5 Line 7**:

“The statistic results show that co-activation of twinning and detwinning occurs in the

intersected NT bundles with $\lambda < 5$ nm under shear stress, and the NTs with $\lambda = 2-3$ nm contribute the highest ratio of 45% (see Supplementary Figure 2)” and **Page 7 Line 13**: “The statistic results exhibit that 80% NTs with $\lambda < 5$ nm (5 observed zones) exhibit the co-activation of twinning and detwinning under shear stress, including intersected and primary NTs. While 100% NTs with $\lambda < 5$ nm (2 observed zones) exhibit detwinning and the subsequent martensitic transformation when the TBs are closely parallel to the tensile direction. Secondary twinning occurs in primary NTs ($\lambda = 6-129$ nm) with a frequency of 80% among the 5 observed zones. Only dislocation motion is observed in the left 2 observed zones, where the λ of those twins are at submicrometer scale. The statistic diagram is given in Supplementary Figure 4.”.

The added supplementary figures (Supplementary Figures 2 and 4.) are given below:

Supplementary Figure 2 Statistic distribution of λ in the intersected NTs to co-activate the twinning and detwinning. Co-activation of twinning and detwinning occurs in the NTs with $\lambda < 5$ nm, and the NTs with $\lambda = 2-3$ nm contribute the highest ratio. The standard deviation of the evaluated λ is 0.5 nm.

Supplementary Figure 4 The statistic diagram of the deformation behaviors. 14 micro-zones are observed through in-situ TEM experiments, where 7 observed zones contain the NTs with $\lambda < 5$ nm, 5 observed zones have the NTs with $5 \text{ nm} < \lambda < 129$ nm, and 2 observed zones are the twins with λ in the submicrometer scale. The standard deviation of the λ are 20 nm for the upper limit of 129 nm and 0.5 nm for the lower limit. The statistic results exhibit that 80% NTs (5 observed zones with $\lambda < 5$ nm) occurs co-activated twinning and detwinning under the 70° angle of TB orientation to loading direction. 100% NTs (2 observed zones with $\lambda < 5$ nm) exhibit detwinning and the subsequent martensite transformation under the 9° angle of TB orientation to loading direction. Among the 5 observed zones containing NTs with $\lambda = 6 - 129$ nm, secondary twinning occurs with a frequency of 80%. In the left 2 observed zones, dislocation motion is active in the NTs with λ in the submicrometer scale.

Reviewer's comment: *1.2. The nucleation of martensite after the detwinning mentioned in the results. Could the authors give some discussion about the internal connection between them? It might also be a good topic to explore in the future work.*

Authors' answer: We do appreciate the reviewer's good suggestion. We add some explanations about the relationship of the martensitic transformation with detwinning, as given in **Page 6 Line 4**: "The detwinning leads to dislocation multiplication by the decomposition of TBs⁴⁰, which induces the early formation of martensitic transformation compared with the other coarse grains (CG)." and in **Page 8 Line 18**: "Accompanied with the detwinning, martensitic transformation is initiated earlier than the other coarse austenite grains, which can further strengthen the materials."

Reviewer's comment: *1.3. The detwinning might lead to the softening in metals. Could the authors add some discussion about the strength-softening effects from different deformation mechanisms?*

Authors' answer: According to the reviewer's suggestion, we add some discussions about the strengthening-softening effects of different deformation mechanisms, as given in **Page 8 Line 11**: "Different from the conventional dislocation strengthening in CGs, the twinning, including the secondary twinning, can simultaneously enhance the strength by dislocation pile-up at TBs and improve the ductility by partials slipping cross TBs^{1,18}. While the detwinning softens the material by the decrease of TBs, which is similar to the effect of grain growth^{6,7}. Thus, the co-activation of

twinning and detwinning can sustain successively work hardening during plastic deformation ²⁸. That is the special deformation behavior of NTs in the very fine scale.”.

Reviewer's comment: *1.4. There are some errors in the reference list like the authors in the reference [2], please re-check that.*

Authors' answer: As kindly mentioned by the reviewer, we correct the reference [2], as shown below:

The reference [2] is changed from “2. Sansoz, F., Lu, K., **Zhu, Y. T.**, & Misra, A. Strengthening and plasticity in nanotwinned metals. *MRS Bulletin* **41**, 292-297 (2016).” into “2. Sansoz, F., Lu, K., **Zhu, T.**, & Misra, A. Strengthening and plasticity in nanotwinned metals. *MRS Bulletin* **41**, 292-297 (2016).”.

Reviewer #2 (Remarks to the Author):

*The authors have addressed some concerns from the previous review. But the referee is still concerned about the novelty of the present paper. The authors stated that essential microstructure characterization of NTed SS has already been reported in their previous work (Chen, A. Y., Ruan, H. H., Wang, J., Chan, H. L., Wang, Q., Li, Q., & Lu, J. The influence of strain rate on the microstructure transition of 304 stainless steel. *Acta Mater.* **59**, 3697-3709 (2011)). They also stated that the mechanical properties of NTed SS have been explored in their previous papers*

(Chen, A. Y., Liu, J. B., Wang, H. T., Lu, J., & Wang, Y. M. Gradient twinned 304 stainless steels for high strength and high ductility. Mater. Sci. Eng. A 667, 179-188 (2016), A.Y. Chen, W.F. Hu, et al., Improving the intergranular corrosion resistance of austenitic stainless steel by high density twinned structure, Scripta Materialia 130 (2017) 264-268). It seems that various deformation mechanisms of NTed SS have also been proposed and discussed in their pervious papers to rationalize the improved mechanical properties of NTed SS. Overall, to this referee, the present paper looks more like an incremental progress instead of an important advance of significance on the understanding of deformation behaviors of NTed SS.

In view of its lack of sufficient novelty, this paper does not meet the standard of Nature Communications. A more specialized journal (e.g., Acta Materialia) might be a better choice.

Authors' answer: In our previous work, we carried out the studies on: (1) How to prepare the multiscale nanotwins and the detailed microstructure of these deformation nanotwins. *(Chen, A. Y., Ruan, H. H., Wang, J., Chan, H. L., Wang, Q., Li, Q., & Lu, J. The influence of strain rate on the microstructure transition of 304 stainless steel. Acta Mater. 59, 3697-3709 (2011)).* (2) The mechanical properties of bulk materials controlled by the gradient nanotwinned structure *(Gradient twinned 304 stainless steels for high strength and high ductility. Mater. Sci. Eng. A 667, 179-188 (2016)).* (3) Corrosion properties of the nanotwinned stainless steel after thermal heat treatment *(A.Y. Chen, W.F. Hu, et al., Improving the intergranular corrosion resistance of austenitic stainless steel by high density twinned structure, Scripta Materialia 130*

(2017) 264-268). In the light of all these related works, we found special properties of nanotwinned materials (high strength and good corrosion resistance), and also meet the difficulties as the other researchers in order to clarify the relationship of microstructure and mechanical properties. The problems are: (1) Strengthening mechanisms exhibit much complicated behaviors, involving the dislocation pile-up at twin boundaries (TBs), hierarchical twinning, TBs migration. (2) All the observed phenomena are based on the post-mortem observations, and confused with the initial structure of deformation nanotwins. Thus, the underlying deformation mechanisms of TBs in nanotwins are still unclear. Therefore, we seek to find out the critical factor and intrinsic mechanisms for guiding the microstructure manipulation.

The objective is achieved in this research work. We directly discover the scale-effect of deformation transition mechanisms of the nanotwinned materials through in-situ TEM experiments, theoretical analysis and molecular dynamics simulations, and built up a fairly comprehensive deformation pattern in the stainless steel. The novelties involve three points: (1) We find two critical scales of the deformation transitions from co-activated twinning/detwinning (twin-lamella-spacing, $\lambda < 5$ nm) to secondary twinning ($5 \text{ nm} < \lambda < 129$ nm), and then to dislocation glide ($\lambda > 129$ nm). That is a creative finding. (2) A full-scale deformation behaviors of nanotwins is mapped as a function of twin-lamella-spacing from nanoscale to microscale according to the direct evidences of in-situ TEM observations. This pattern is very helpful for understanding the deformation behaviors in nanotwinned metals and designing advanced structural materials with excellent mechanical properties. (3)

We reveal the intrinsic mechanisms in deformation nanotwins to govern the deformation transitions by establishing a dislocation-based theoretical model and molecular dynamics simulations. All these findings are newly discovered.

Reviewer #3 (Remarks to the Author):

The authors have addressed most of my comments. From their response, I realized why I was so confused about the way they defined the deformation map based on twin boundary spacing. Set by the literature work on nanotwin materials, I have the tendency to think each tested sample should contain only one region. Instead, it seems that the sample in this work can contain twins with much different sizes. The deformation mechanism of different regions in this sample depends on the twin spacing in this region. In other words, a nanotwinned sample can deform simultaneously through all of the mechanism proposed in this work. If my understanding is right, the authors should make this point clear in the abstract to avoid this kind of confusion to the prospective readers.

Authors' answer: The understanding of the reviewer is right. The specimen prepared by surface mechanical attrition treatment is composed of deformation nanotwins with wide twin-spacing sizes from several nanometers to hundreds of nanometers. During in-situ tensile, the different observed regions contain different sizes of deformation nanotwins, and exhibit different deformation behaviors. As the suggestion of the reviewer, we add the corresponding content in the abstract, as given in following:

“Tensile characteristics of the deformation NTs are in-situ observed under transmission electron microscope via different regions in the specimen containing NTs with twin-lamella-spacing (λ) from several nanometers to hundreds of nanometers.”.

We are very grateful to both the editor and the referees for their very helpful suggestions on this manuscript.

Best regards!

A.Y. Chen

REVIEWERS' COMMENTS:

Reviewer #1 (Remarks to the Author):

The authors revised the part suggested before. The mechanism transition with the twin spacing change mainly originated from the competition between the glides of partial and full dislocations and this concept is still not very novel, although the authors emphasized that the construction of the deformation map is a key issue. Therefore, I suggest this work to be published in other specialized journals.

Response to the Reviewers' Comments

Ref. No.: NCOMMS-18-14200C

Title: Scale law of complex deformation transitions of nanotwins in stainless steel

Authors: A. Y. Chen, L. L. Zhu, L. G. Sun, J. B. Liu, H. T. Wang, X. Y. Wang, J. H. Yang, J. Lu

REVIEWERS' COMMENTS:

Reviewer #1 (Remarks to the Author):

The authors revised the part suggested before. The mechanism transition with the twin spacing change mainly originated from the competition between the glides of partial and full dislocations and this concept is still not very novel, although the authors emphasized that the construction of the deformation map is a key issue. Therefore, I suggest this work to be published in other specialized journals.

Authors' answer: Many thanks for the reviewer to deepen the understanding of our work. The deformation mechanisms of the nanotwins are very important for the structure design and manipulation for the development of advanced high strength steel, and therefore the twin-spacing-effect on the strengthening becomes the critical factor.

In this work, we find two novel deformation transitions depending on the

twin-lamella-spacing (λ) by in-situ TEM observations, that is co-activated twinning/detwinning ($\lambda < 5$ nm) to secondary twinning ($5 \text{ nm} < \lambda < 129$ nm), and then to dislocation glide ($\lambda > 129$ nm). On basis of these observations, we construct a clear deformation pattern of NTs with twin-lamella-spacing varying from several nanometers to hundreds of nanometers by combining in-situ tensile tests with a dislocation-based theoretical model and molecular dynamics simulations. Although we discuss the effects of the types of partial dislocations (not involving in full dislocations) and the TB orientation on the deformation transition mechanisms in order to verify the critical limits in the simulations, we deepen on the understanding of pre-set partial dislocations in deformation nanotwins, and these results are the extended results.

Although the nanotwinned phenomenon have been observed in many model materials, the concept of the construction of deformation map is a new and important paradigm to better understand and design nanotwinned engineering materials. We consider that it's an important step to integrate this new method for the real engineering materials development. For example, many newly-developed high entropy alloys have a low stacking faults energy that may potentially subject to the need of a such concept for further enhancements of mechanical properties using this nanotwinned engineering concept.

We appreciate the time of the editor and reviewers that has been spent in reading our manuscript and their constructive comments on this manuscript.

Best regards!

A.Y. Chen